**Cite this article:** de los Reyes V AA, Kim Y. 2022 Optimal regulation of tumour-associated neutrophils in cancer progression. *R. Soc. Open Sci.* **9**: 210705.

computational biology/mathematical modelling

tumour-associated neutrophils, TGF-beta, IFN-beta, optimal control, tumour growth, mathematical model

**Author for correspondence:**
Yangjin Kim
e-mail: ahyouhappy@gmail.com

# Optimal regulation of tumour-associated neutrophils in cancer progression

Aurelio A. de los Reyes V[1,2] and Yangjin Kim[3,4]

 

[1]Biomedical Mathematics Group, Pioneer Research Center for Mathematical and Computational Sciences, Institute for Basic Science, Daejeon 34126, Republic of Korea
[2]Institute of Mathematics, University of the Philippines Diliman, Quezon City 1101, Philippines
[3]Department of Mathematics, Konkuk University, Seoul 05029, Republic of Korea
[4]Mathematical Biosciences Institute, Columbus, OH 43210, USA

AAdIRV, 0000-0001-5418-4579; YK, 0000-0002-8905-8481

In a tumour microenvironment, tumour-associated neutrophils could display two opposing differential phenotypes: anti-tumour (N1) and pro-tumour (N2) effector cells. Converting N2 to N1 neutrophils provides innovative therapies for cancer treatment. In this study, a mathematical model for N1-N2 dynamics describing the cancer survival and immune inhibition in response to TGF-$\beta$ and IFN-$\beta$ is considered. The effects of exogenous intervention of TGF-$\beta$ inhibitor and IFN-$\beta$ are examined in order to enhance N1 recruitment to combat tumour progression. Our approach employs optimal control theory to determine drug infusion protocols that could minimize tumour volume with least administration cost possible. Four optimal control scenarios corresponding to different therapeutic strategies are explored, namely, TGF-$\beta$ inhibitor control only, IFN-$\beta$ control only, concomitant TGF-$\beta$ inhibitor and IFN-$\beta$ controls, and alternating TGF-$\beta$ inhibitor and IFN-$\beta$ controls. For each scheme, different initial conditions are varied to depict different pathophysiological condition of a cancer patient, leading to adaptive treatment schedule. TGF-$\beta$ inhibitor and IFN-$\beta$ drug dosages, total drug amount, infusion times and relative cost of drug administrations are obtained under various circumstances. The control strategies achieved could guide in designing individualized therapeutic protocols.

## 1. Introduction

Cancer is the leading cause of death worldwide [1,2]. Various immune cells including neutrophils [3], natural killer (NK) cells [4] and macrophages [5] in tumour microenvironment (TME) play a major role in regulation of tumour growth and

anti-tumour treatment in many cancers including lung cancers. Copious neutrophils in the blood constitute the first protection in innate immunity [6,7]. Approximately $10^{11}$ neutrophils are generated in the bone marrow and released into the blood circulation every day [8]. Neutrophil recruitment at sites of infection or injury is triggered by the release of pathogen- or damage-associated molecular patterns from invading microorganisms or damaged and/or dead cells, respectively [9]. The process of neutrophil extravasation comprises a complex multistep cascade that is tightly regulated by coordinated sequence of adhesive and migratory events [6,10]. Neutrophils undergo a series of mechanisms, including phagocytosis, and the development and regulation of neutrophil extracellular traps (NETs) [11]. Furthermore, neutrophils release proteinases into the surrounding tissue damaging the host [12], and produce cytokines and chemokines influencing inflammatory and immune responses [13,14].

Aside from the traditional antimicrobial functions, there is increasing evidence suggesting that tumour-associated neutrophils (TANs) play a major role in tumour progression from formation to malignant state [15–19]. Several clinical and laboratory studies have indicated that the presence of TANs has been assessed with poor prognosis in various tumours including metastatic melanoma [20], bronchoalveolar carcinoma [21], renal carcinoma [22], and head and neck squamous cell carcinoma (HNSCC) [23]. In these conditions, neutrophils exhibit a pro-tumour phenotype which is detrimental to the host. TME controls neutrophil recruitment and thus, TANs promote tumour progression. Hence, TANs display two differential phenotypes [15,16,24]: (i) an anti-tumourigenic role (called N1), and (ii) pro-tumour progression (called N2).

There is mounting evidence that the immunosuppressive cytokine TGF-$\beta$ skews neutrophil differentiation towards N2 phenotype [24–26], while TGF-$\beta$ inhibitor and type-1 IFN ($\alpha$, $\beta$, $\omega$) therapy are recognized to alter neutrophils toward the N1 phenotype [27,28]. Several studies have shown that IFN-$\beta$ in tumour microenvironment interacts with key molecules in signalling pathways activating its functions to stimulate anti-tumour activities [29–32]. Therefore, normalization of TME by the N2→N1 transition can lead to better control of tumour growth. For example, injection of IFN-$\beta$ [29,31,33–35] or TGF-$\beta$ inhibitor such as galunisertib [36–40] and LY2109761 [41,42] can be a novel anti-tumour strategy due to their effective tumour killing and anti-tumour immune response [43,44].

Mathematical modelling is a valuable tool in providing information to understand the complex mechanisms and processes in cancer [45,46]. Models can generate theories and hypotheses on quantitative grounds and provide reasonable predictions that could aid scientists towards the next series of experiments [47]. Furthermore, mathematical modelling can help clinicians to evaluate tumour microenvironment and develop anti-tumour strategies [48–52]. In addition, in silico models could play a significant role in optimizing clinical trial design and patient stratification, suggesting different treatment modalities, and alter risk prognoses [53–59].

In this work, we consider the following variables in a mathematical model:

$C(t)$ = density of the N2 complex at time $t$;
$I(t)$ = density of the N1 complex at time $t$;
$T(t)$ = tumour volume at time $t$;
$G(t)$ = concentration of TGF-$\beta$ at time $t$;
$L(t)$ = concentration of TGF-$\beta$ inhibitor at time $t$;
$S(t)$ = concentration of IFN-$\beta$ at time $t$;

In this study, regulation of N1 and N2 TANs is explored through TGF-$\beta$ inhibitor and IFN-$\beta$ controls in the framework of optimal control theory having the objective of minimizing the tumour size with least drug administration cost. In particular, infusion scheme(s) accounting for the amount and frequency of administration is explored. Various initial N1 and N2 phenotypes are also considered to simulate individualized therapeutic regimen depending on current pathological conditions. Several control strategies including combination therapies are examined under different circumstances that could give the best clinical outcomes.

We found therapeutic regimen at regulating anti- and pro-tumoural neutrophil phenotypic states by means of four different administration modalities of a TGF-$\beta$ inhibitor and IFN-$\beta$ cytokine in lung cancer. The optimal control strategy predicts that, depending on pathological states, therapies may have to be adjusted accordingly to minimize adverse effects of drugs and its administration cost.

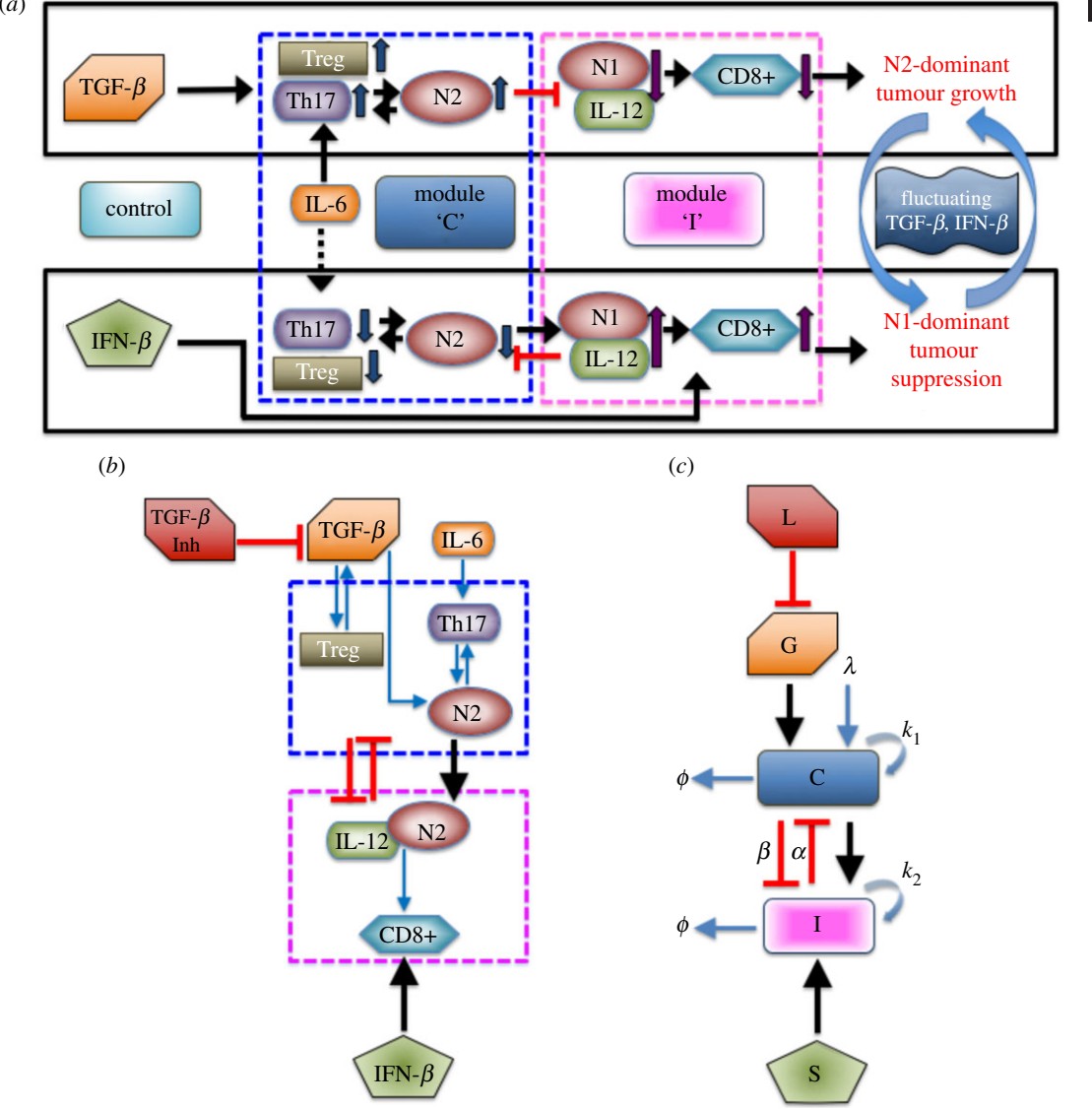

**Figure 1.** (a) N1 and N2 complexes regulate cancer survival and immune inhibition in the presence of TGF-$\beta$ and IFN-$\beta$ [60]. (b) Simplified model diagram of N1-N2 network dynamics with the inclusion of a TGF-$\beta$ inhibitor acting as a control to the signalling pathway. (c) Schematic non-dimensional model including a TGF-$\beta$ inhibitor (modified from [60]). Optimal control is applied to obtain appropriate L and S dosages that minimize the tumour volume with minimal costs.

## 2. Methodology

### 2.1. Mathematical model

In order to understand the mutual interactions between a tumour and immune system involving tumour-associated neutrophils, a mathematical model of the signalling pathway for lung cancer is proposed in [60]. It consists of the basic network of interactions between cells, cytokines and growth factors. It was shown that IFN-$\beta$ and TGF-β play a crucial function as tumour suppressor and tumour promoter (refer to figure 1a). High levels of TGF-$\beta$ and low activities of IFN-$\beta$ induce Treg infiltration generating inhibition of overall anti-tumour immune response of T cells. In addition, TGF-β alters the tumour microenvironment inducing polarization of neutrophil N1 into N2 type while IFN-$\beta$ can polarize the N2 phenotype back to N1 phenotype [15,61]. The N2-dominant microenvironment contributes to enhance tumour growth. On the contrary, low TGF-$\beta$ and high IFN-$\beta$ levels reduce the neutrophil polarization toward N2 from N1 phenotype and enhance immune activities of T cells and N1 TANs, driving tumour suppression. For model simplification, the N2 regulatory network among Treg, Th17 and N2 cells is merged into one

**Table 1.** Parameters used in the model.

| parameter | description | value | references |
|-----------|-------------|-------|------------|
| $\lambda$ | source (IL-6) term in the N2 component | 0.01 | [60,62] |
| $G$ | source (TGF-$\beta$) term in the N2 component | 0–1.0 | [60,63] |
| $k_1$ | self-regulation rate in the N2 component | 4.0 | [60] |
| $k_3$ | scaling parameter | 1.0 | [60] |
| $\alpha$ | suppression rate of the N2 component by the N1 component | 1.5 | [60] |
| $k_2$ | self-regulation rate in the N1 component | 4.0 | [60] |
| $k_4$ | scaling parameter | 1.0 | [60] |
| $\beta$ | suppression rate of the N1 component by the N2 component | 1.0 | [60] |
| $S$ | source (IFN-$\beta$) term in the N1 component | 0.2 | [60,64] |
| $\mu$ | decay rate of the N1 component | 1.0 | [65,66] |
| $C_{th}$ | threshold (N2 component) | 1.81 | [60] |
| $I_{th}$ | threshold (N1 component) | 1.29 | [60] |
| $r$ | growth rate of tumour cells | 0.05 | [33] |
| $K$ | scaling constant | 1.0 | [33] |
| $\gamma_1$ | suppression strength in immune-mediated growth | 0.1 | [33] |
| $T_0$ | carrying capacity of cancer cells | 100 | [33] |
| $G_S$ | source term in the TGF-$\beta$ component | 0.826 | [60] |
| $\mu_G$ | decay rate (TGF-$\beta$) | 0.826 | [5,67–69] |
| $\gamma_L$ | degradation rate of TGF-$\beta$ through the inhibitor | 100 | [60] |
| $\mu_L$ | decay rate (TGF-$\beta$ inhibitor) | 6.6 | [36] |
| $\mu_S$ | decay rate (IFN-$\beta$) | 3.96 | [70] |

component denoted as N2 complex. On the other hand, the immunoregulatory system composed of N1 cells, interleukin IL-12 and CD8+, is incorporated into N1 complex. Further details of model development can be found in [60]. The associated dimensionless equations for complexes N2 ($C$), N1 ($I$) and tumour volume ($T$) is described by the following ordinary differential equations [60]:

$$
\begin{aligned}
\frac{\mathrm{d}C}{\mathrm{d}t} &= \underbrace{\lambda}_{\text{source (IL-6)}} + \underbrace{\lambda_G G}_{\text{source (TGF-}\beta)} + \underbrace{\frac{k_1}{k_3^2 + \alpha I^2}}_{\text{inhibition from N1}} - \underbrace{C}_{\text{decay}}, \\
\frac{\mathrm{d}I}{\mathrm{d}t} &= \underbrace{\lambda_S S}_{\text{source (IFN-}\beta)} + \underbrace{\frac{k_2}{k_4^2 + \beta C^2}}_{\text{inhibition from N2}} - \underbrace{\mu I}_{\text{decay}}
\end{aligned}
\right\}
\tag{2.1}
$$

and

$$
\frac{\mathrm{d}T}{\mathrm{d}t} = \underbrace{r\left(1 + \frac{C}{K + \gamma_1 I}\right)T\left(1 - \frac{T}{T_0}\right)}_{\text{growth}} - \underbrace{\delta I T}_{\text{killing}},
$$

with essential set of parameters listed in table 1. Here, two sources (IL-6, TGF-$\beta$) of the N2 complex are represented in the first and second terms in equation of $C$ while the IFN-$\beta$-mediated source of the N1 complex is provided in the first term in equation of $I$. The third term in equation of $C$ and second term in equation of $I$ represent the mutual inhibition between N1 and N2, respectively. Tumour growth and N1-mediated tumour cell killing are represented in the first and second terms in equation of $T$, respectively. Finally, the decay process of the N1 and N2 complexes is represented in the last terms in equations of $I$ and $C$, respectively.

The basic dynamics of the model (2.1) is illustrated in figure 2. Different levels of TGF-$\beta$ ($G$) signals could induce either a single stable steady state SS$^{(s)}$ or three steady states of which two are stable and one

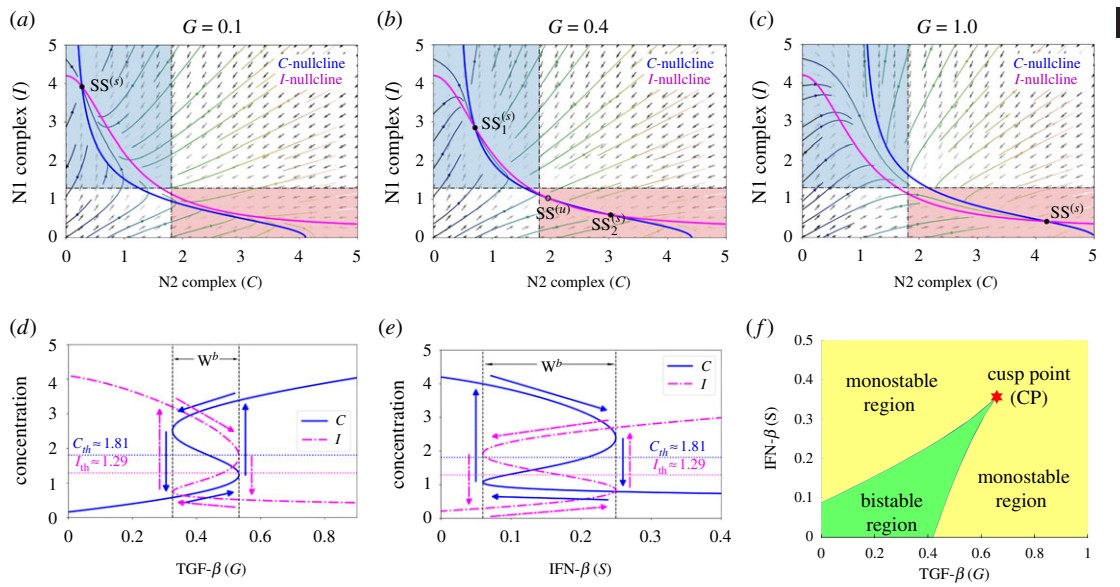

**Figure 2.** ($a$–$c$) Nullclines of model (2.1) in the $C$–$I$ phase plane in response to ($a$) low, $G = 0.1$, ($b$) intermediate, $G = 0.4$, and ($c$) high, $G = 1$ levels of TGF-$\beta$ showing the corresponding steady states where SS$^{(s)}$ and SS$^{(u)}$ denote stable, respectively, unstable steady state. Anti-tumorigenic and tumorigenic regions are marked in blue and pink boxes in ($a$–$c$). ($d,e$) Hysteresis diagram with respect to varying signals of TGF-$\beta$ and IFN-$\beta$ promoting N1-N2 on–off switch activation. ($f$) Codimension 2 bifurcation for different $G$ and $S$ levels depicting division of bistable and monostable region and a cusp point (CP).

unstable SS$^{(u)}$. Here, $S$ represents the IFN-$\beta$-induced source of the N1 module. When TGF-$\beta$ is low ($G = 0.1$), N2 complex activities are suppressed while N1 activities are promoted yielding one SS$^{(s)}$ which induces an anti-tumorigenic phenotype of TANs (figure 2$a$). On the contrary, high level of TGF-$\beta$ ($G = 1.0$) enhances N2 activities but decreases that of N1 complex leading to a stable SS$^{(s)}$ which is a tumorigenic phenotype of TANs (figure 2$c$). Thus, for low and high $G$ signals, phenotypic differentiation of TANs does not depend on initial status of N1 and N2 complexes. However, an intermediate TGF-$\beta$ signal ($G = 0.4$) leads to two SS$^{(s)}$ and one SS$^{(u)}$ generating bistability in the system. The initial conditions of N1 and N2 complex influence the induction of either anti- or pro-tumour TANs (figure 2$b$). Due to the dichotomous behaviour of TANs with respect to signals of TGF-$\beta$, an associated hysteresis diagram is expected (figure 2$d$). The bistability window $W^b \approx [0.33, 0.53]$ provides the limit points for which the system jumps from one stable state to the other. The system is not only $G$-dependent but also $S$-dependent. That is, varying levels of IFN-$\beta$ signals also promotes bistability and hysteresis as shown in figure 2$e$, where the bistability window with respect to $S$ is $W^b \approx [0.06, 0.25]$. Note that the domain of bistability is dependent on key parameters and may progressively shrink for some parameter sets. Figure 2$f$ depicts the bistable and monostable region under varying $G$ and $S$ simultaneously. Starting at a limit point obtained from one-parameter bifurcation ($G$), a cusp point (CP) is achieved by computing a fold continuation. Then, two key parameters $G$ and $S$ are increased (or decreased) at the same time along the trajectory of a limit point for equilibrium curve at the corresponding value of $S$. This resulted to a codim 2 bifurcation. The cusp point yields the threshold values $C_{th}$ and $I_{th}$. `MatCont` was used for numerical continuation and bifurcation analysis [71]. In this study, the effects of exogenous intervention of TGF-$\beta$ inhibitor and IFN-$\beta$ on tumour progression are further investigated. Figure 1$a$ illustrates the response of N1 and N2 complexes to the TGF-$\beta$ inhibitor and IFN-$\beta$, acting as *regulatory controls* to the system. Note that N2-polarization by TGF-$\beta$ can be blocked by using inhibitor (TGF-$\beta$ Inh) and/or IFN-$\beta$ promoting N1-dominant microenvironment. As depicted, both controls could downregulate N2 complex which activates N1 complex leading to tumour-suppression. Note that IL-6 is a protein-coding gene that induces Treg and others (upward arrow), which promotes N2 phenotype resulting in tumour growth. The dashed arrow indicates that withdrawal of IL-6 activity downregulates Treg and others (hence, downward arrow), leading to N1 phenotype suppressing tumours. Figure 1$b$ shows the schematic diagram of the simplified model while figure 1$c$ is the dimensionless form; N1 and N2 complexes are grouped as module $C$ and $I$, respectively.

The non-dimensional set of differential equations describing the model is as follows:

$$
\left.
\begin{aligned}
\frac{dL}{dt} &= u_L(t) - \mu_L L, \\
\frac{dG}{dt} &= G_S - \mu_G G - \gamma_L LG, \\
\frac{dC}{dt} &= \lambda + \lambda_G G + \frac{k_1}{k_3^2 + \alpha I^2} - C, \\
\frac{dS}{dt} &= u_S(t) - \mu_S S, \\
\frac{dI}{dt} &= \lambda_S S + \frac{k_2}{k_4^2 + \beta C^2} - \mu I \\
\frac{dT}{dt} &= r\left(1 + \frac{C}{K + \gamma_1 I}\right) T\left(1 - \frac{T}{T_0}\right) - \delta I T,
\end{aligned}
\right\}
\tag{2.2}
$$

and

where $u_L$ and $u_S$ denote the exogenous infusion rate of TGF-$\beta$ inhibitor and IFN-$\beta$, respectively, and $T$ is the tumour volume.

The current modelling approach uses *optimal control theory* to identify therapeutic infusion protocol of TGF-$\beta$ inhibitor and IFN-$\beta$ that could lead to N1-polarization suppressing tumour progression. In particular, we aim to determine appropriate drug injection rates $u_L(t)$ and/or $u_S(t)$ minimizing tumour volume and associated administrative infusion cost. The controls $u_L$ and $u_S$ represent dosages of TGF-$\beta$ inhibitor and an IFN-$\beta$ drug, respectively. These controls are assumed to be bounded and Lebesgue integrable. That is, the control set is defined as

$$
\Omega = \{(u_L, u_S) \big| u_i \quad \text{is measurable with } 0 \leq u_i(t) \leq u_i^{\max}, \ t \in [t_0, t_1], \ i = L, S\}, \tag{2.3}
$$

where $u_L^{\max}$ and $u_S^{\max}$ are the maximum tolerated TGF-$\beta$ inhibitor and an IFN-$\beta$ drug doses, respectively, which could be administered in a given treatment period. These upper bounds may also be viewed as the maximum amounts a patient can financially afford. The lower bounds for $u_L$ and $u_S$ correspond to no treatment. Furthermore, we intend to administer precisely the amount of treatment to be given to the patient over the given time interval $[t_0, t_1]$, which is within safety limits (*isoperimetric constraint*). That is, we have

$$
\int_{t_0}^{t_1} u_L(t) \, dt = A_1 \quad \text{and} \quad \int_{t_0}^{t_1} u_S(t) \, dt = A_2, \tag{2.4}
$$

where $A_1$ and $A_2$ are the known TGF-$\beta$ inhibitor and IFN-$\beta$ drug amounts, respectively. Considering the bounds for $u_L$ and $u_S$, the choices for $A_1$ and $A_2$ satisfy

$$
0 \leq A_1 \leq u_L^{\max}(t_1 - t_0) \quad \text{and} \quad 0 \leq A_2 \leq u_S^{\max}(t_1 - t_0).
$$

The following optimal control problems (OCPs) are formulated to examine different therapeutic strategies. Here, the control costs are modelled by linear combination of quadratic terms, where the weight factors $B_1$ and $B_2$ denote the relative cost of minimizing tumour volume $T(t)$ and administering TGF-$\beta$ inhibitor and/or IFN-$\beta$ drug infusions over a certain time period, respectively. The quadratic form of the cost functional with respect to the controls ensures that the associated Hamiltonian becomes strictly convex and has a unique minimizer, thus making the mathematical problem more tractable. The use of such quadratic functionals is not commonly motivated by biological phenomena making its simplifying assumptions questionable [72]. However, its use in order to incorporate the nonlinearity of the problem has produced qualitatively significant results [73,74]. It should be noted that the quadratic controls in the cost function model the adverse effects of using too much drugs, as used in several works [73,75–79]. An isoperimetric optimal control has been applied to cancer immunotherapy [80–82].

1. *TGF-$\beta$ inhibitor control only*. In this scheme, we want to investigate the anti-tumour effect of TGF-$\beta$ inhibitor on tumour growth. In particular, we want to minimize the tumour size ($T$) and dose of TGF-$\beta$ inhibitor while keeping the same IFN-$\beta$ supply. Thus, the OCP is to minimize the cost

functional

$$J\big(u_{\mathrm{L}}(t)\big) = \int_{t_0}^{t_1} \left( T(t) + \frac{B_1}{2} u_{\mathrm{L}}^2(t) \right) \mathrm{d}t, \tag{2.5}$$

subject to (2.2) where $S$ is assumed constant (i.e. $S = 0.2$, $\mathrm{d}S/\mathrm{d}t = 0$) with $\int_{t_0}^{t_1} u_{\mathrm{L}}(t)\,\mathrm{d}t = A_1$.

2. *IFN-β control only*. We consider the anti-tumour efficacy of IFN-β infusion. In particular, we want to minimize the tumour size ($T$) and dose of IFN-β in the absence of IFN-β inhibitor. Thus, the problem is formulated as minimizing the cost functional

$$J\big(u_{\mathrm{S}}(t)\big) = \int_{t_0}^{t_1} \left( T(t) + \frac{B_2}{2} u_{\mathrm{S}}^2(t) \right) \mathrm{d}t, \tag{2.6}$$

subject to (2.2) with constant TGF-β and no inhibitor present (i.e. $G = 0.45$, $\mathrm{d}G/\mathrm{d}t = 0$, $L = 0$, $\mathrm{d}L/\mathrm{d}t = 0$) with $\int_{t_0}^{t_1} u_{\mathrm{S}}(t)\,\mathrm{d}t = A_2$.

3. *Concomitant TGF-β inhibitor and IFN-β controls*. In this approach, the anti-tumour efficacy of both controls is examined assuming that TGF-β inhibitor and IFN-β can be administered simultaneously. Specifically, we want to minimize the tumour size ($T$) and doses of both TGF-β inhibitor and IFN-β. Hence, the OCP is to minimize

$$J\big(u_{\mathrm{L}}(t), u_{\mathrm{S}}(t)\big) = \int_{t_0}^{t_1} \left( T(t) + \frac{B_1}{2} u_{\mathrm{L}}^2(t) + \frac{B_2}{2} u_{\mathrm{S}}^2(t) \right) \mathrm{d}t, \tag{2.7}$$

subject to (2.2) and isoperimetric constraints (2.4).

4. *Alternating TGF-β inhibitor and IFN-β controls*. In order to avoid drug complications such as risk of killing healthy cells and overdose, we propose a strategy of administering drugs alternately. Specifically, we want to minimize the tumour size ($T$) and doses of both TGF-β inhibitor and IFN-β in an alternating injection scheme. Thus, the OCP can then be formulated as minimizing the cost functional

$$\begin{aligned} J\big(u_{\mathrm{L}}(t), u_{\mathrm{S}}(t)\big) = {} & \int_{t_0}^{t_1} \left( T(t) + \frac{B_1}{2} u_{\mathrm{L}}^2(t) \right) \mathrm{d}t \\ & + \int_{t_0'}^{t_1'} \left( T(t) + \frac{B_2}{2} u_{\mathrm{S}}^2(t) \right) \mathrm{d}t \end{aligned} \tag{2.8}$$

subject to (2.2) and isoperimetric constraints (2.4) within an appropriate time interval.

Our goal is to find optimal infusion regimen for TGF-β inhibitor and/or IFN-β, $\mathbf{u}^*(t) = (u_{\mathrm{L}}^*(t), u_{\mathrm{S}}^*(t))$ such that

$$J\big(\mathbf{u}^*(t)\big) = \min_{\Omega} J\big(\mathbf{u}(t)\big), \tag{2.9}$$

where $\Omega$ is the set of all square integrable functions $u_{\mathrm{L}}(t)$ and $u_{\mathrm{S}}(t)$ over a specified interval given in (2.3). By using standard results in control theory, the existence of optimal controls is guaranteed [83]. In this optimal control problem (OCP), the integrand of the objective functional satisfies the necessary convexity, in which Pontryagin's maximum principle can be applied [84]. In solving the OCP, an iterative method is used where initial conditions for the state variables and terminal conditions for the adjoints constitute a two-point boundary value problem. State equations are solved forward in time using the given initial conditions and the corresponding adjoint equations are solved backward in time starting with the transversality conditions. This is known as the forward–backward sweep method (FBSM). Numerical simulations can be obtained using Euler scheme or fourth-order iterative Runge–Kutta method. We use adapted FBSM [85,86] with adapted scheme [87] for isoperimetric constraints.

# 3. Results and discussion

In this section, we present the numerical results obtained from different control strategies. The weight parameters $B_1 = B_2 = 1$ are used as default values. Four different initial conditions, $Q_i := (C_{0,i}, I_{0,i})$, $i = 1$, …, 4, are considered in order to resemble diverse phenotypical state of N1 and N2 complexes. The polarization of the microenvironment is N2-dominant if $C > C_{\mathrm{th}}$ and $I < I_{\mathrm{th}}$, otherwise, N1-dominant when $C < C_{\mathrm{th}}$ and $I > I_{\mathrm{th}}$. We include the following conditions:

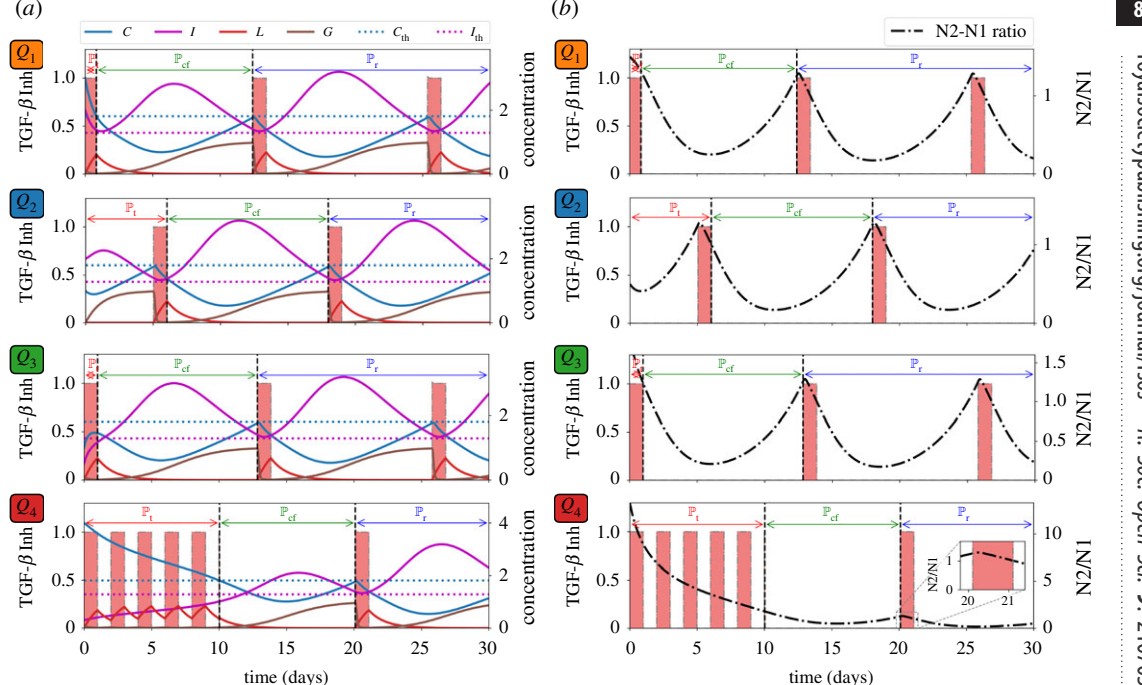

**Figure 3.** TGF-$\beta$ inhibitor administration schedule under different initial conditions $Q_1$, $Q_2$, $Q_3$, $Q_4$ and corresponding dynamics of (a) $C$, $I$, $L$, $G$, and (b) N2/N1 levels. Administration of TGF-$\beta$ inhibitor decreases the N2/N1 levels.

— $Q_1 = (3, 2)$: low-risk state, high levels of both $C$ and $I$ above their corresponding threshold values;

— $Q_2 = (1, 2)$: risk-free state, low (high) level of $C$ ($I$) below (above) the threshold value;

— $Q_3 = (1, 0.5)$: low-risk state, low levels of $C$ and $I$ below their corresponding threshold values; and

— $Q_4 = (4, 0.3)$: high-risk state, high (low) level of $C$ ($I$) above (below) the threshold value.

An important phenotypic marker is

$$C < C_{th} \quad \text{and} \quad I > I_{th}. \tag{3.1}$$

## 3.1. TGF-$\beta$ inhibitor control only

In this therapeutic strategy, we assume that there is a constant source of IFN-$\beta$ and that the system can be regulated by exogenously administering TGF-$\beta$ inhibitor. The goal is to obtain optimal infusion protocol with the least cost of TGF-$\beta$ inhibitor infusion, i.e. $u_L^*(t)$ that represses the N2-dominant phenotype leading to tumour-suppression. We consider that the total dose of TGF-$\beta$ inhibitor which can be administered in a day is 1 unit, and thus, $A_1 = 1$. Also, we presume that this drug can only be administered every other day as in experiments [37] taking into account its toxicity to healthy tissue. The adapted FBSM is applied for (a maximum of) one day if condition (3.1) is not satisfied (i.e. either $C > C_{th}$ or $I < I_{th}$ or both inequalities hold) to obtain the optimal control profile $u_L^*(t)$ for that period. It is expected that $C$ will decrease since TGF-$\beta$ inhibitor blocks the TGF-$\beta$ promoting N2 phenotype which would in turn, upregulate N1 activities. Then, a day of no infusion follows letting the TGF-$\beta$ inhibitor take effect. The activities of $C$ and $I$ are simultaneously monitored. At the time when condition (3.1) holds, infusion is stopped or no control is applied. Continuous monitoring of $C$ and $I$ profiles will be crucial in determining the next TGF-$\beta$ inhibitor infusion. This routine suggests a TGF-$\beta$ inhibitor control.

We consider the *treatment period* $\mathbb{P}_t$ to be the time when initial control infusion is administered until condition (3.1) fails. Then *cancer-free period* $\mathbb{P}_{cf}$ is shortly achieved when condition (3.1) is satisfied. Before $C$ crosses $C_{th}$ from below ($I$ crosses $I_{th}$ from above), optimal control is again applied proposing the specific time of TGF-$\beta$ inhibitor administration, and that commences the *relapse period* $\mathbb{P}_r$. Figure 3a depicts the control scheme, the corresponding profile of TGF-$\beta$ inhibitor ($L$) and its effect on TGF-$\beta$ ($G$), and the concentration profiles of $C$ and $I$ with different initial conditions $Q_1$, $Q_2$, $Q_3$, $Q_4$. For instance, if one considers $Q_4$ as initial values of $C$ and $I$, TGF-$\beta$ inhibitor control is applied every other

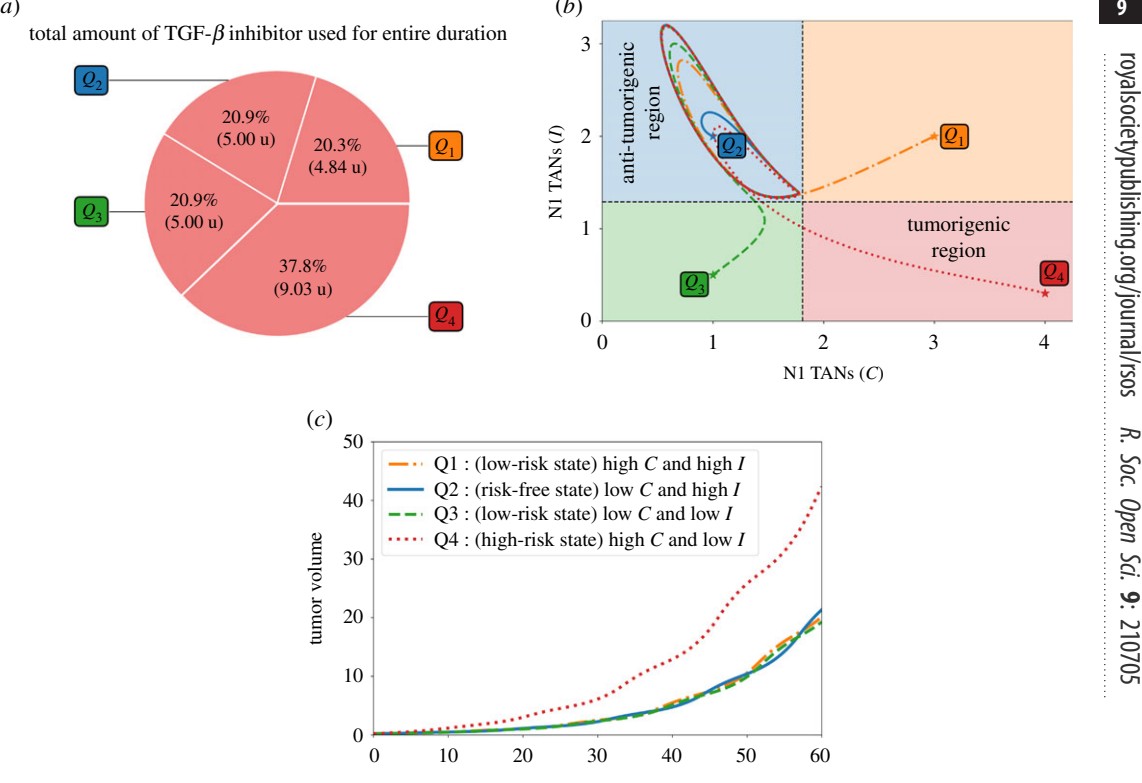

**Figure 4.** (a) Proportion of the amount of TGF-$\beta$ inhibitor used for the entire treatment duration, (b) dynamics of the N1-N2 system and (c) tumour dynamics under TGF-$\beta$ inhibitor control starting at different initial conditions $Q_1$, $Q_2$, $Q_3$, $Q_4$.

day until $C$ crosses $C_{th}$ which is about 10 days, duration of $\mathbb{P}_t$. Then $\mathbb{P}_{cf}$ follows until around day 20 when TGF-$\beta$ inhibitor administration is again needed. The remaining time period covers $\mathbb{P}_r$. Note also that starting at risk-free, anti-tumorigenic region $Q_2$ satisfies condition (3.1). Thus, initial TGF-$\beta$ inhibitor infusion commences (later) at around day 5. Figure 3b illustrates that administration of TGF-$\beta$ inhibitor is needed to reduce the N2/N1 ratio (shown in black, dash-dotted curve). It is expected that when the initial condition is at high-risk, tumorigenic region $Q_4$, the total amount of TGF-$\beta$ inhibitor to be used for entire medication period (60 days) is the largest. This is shown in figure 4a. The $C-I$ dynamics obtained from the TGF-$\beta$ inhibitor control protocol at different initial conditions are illustrated in figure 4b. Observe that all the trajectories converge to the anti-tumorigenic region. Time courses of tumour volume in these four cases illustrate the relatively poor anti-tumour efficacy when the initial condition is in the tumorigenic region $Q_4$ (figure 4c), despite the highest cost (figure 4a).

## 3.2. IFN-$\beta$ control only

Let us consider a scenario when TGF-$\beta$ inhibitor cannot be administered due to adverse drug reactions, high cost, etc., and assume that only IFN-$\beta$ can be used for treatment purposes. Taking into account that IFN-$\beta$ is a noxious drug [29,88,89], it can be administered only every other day in increasing amount as illustrated in experimental system [29,33,44]. In this strategy, we let the minimum and maximum IFN-$\beta$ amount be 2 and 10 units, respectively (i.e. $A_2^{min} = 2$, $A_2^{max} = 10$). As in the first control strategy, both $C$ and $I$ are assessed. If condition (3.1) fails, then immediate administration of IFN-$\beta$ is needed. This is done by using adapted FBSM for a day to fully utilize the minimum amount followed by a period of no infusion (maximum of a day) where both $C$ and $I$ are concurrently monitored. If condition (3.1) still does not hold, application of control routine is carried out where IFN-$\beta$ amount is increased by 2 units. This infusion scheme with increasing amount of IFN-$\beta$ per administration is implemented until condition (3.1) is not satisfied. The period from first IFN-$\beta$ control administration until the last infusion where at least one of the inequalities in condition (3.1) no longer holds is what we refer to as the *treatment period* $\mathbb{P}_t$. Then a *critical period* $\mathbb{P}_{crit}$ follows where both $C$ and $I$ should be carefully

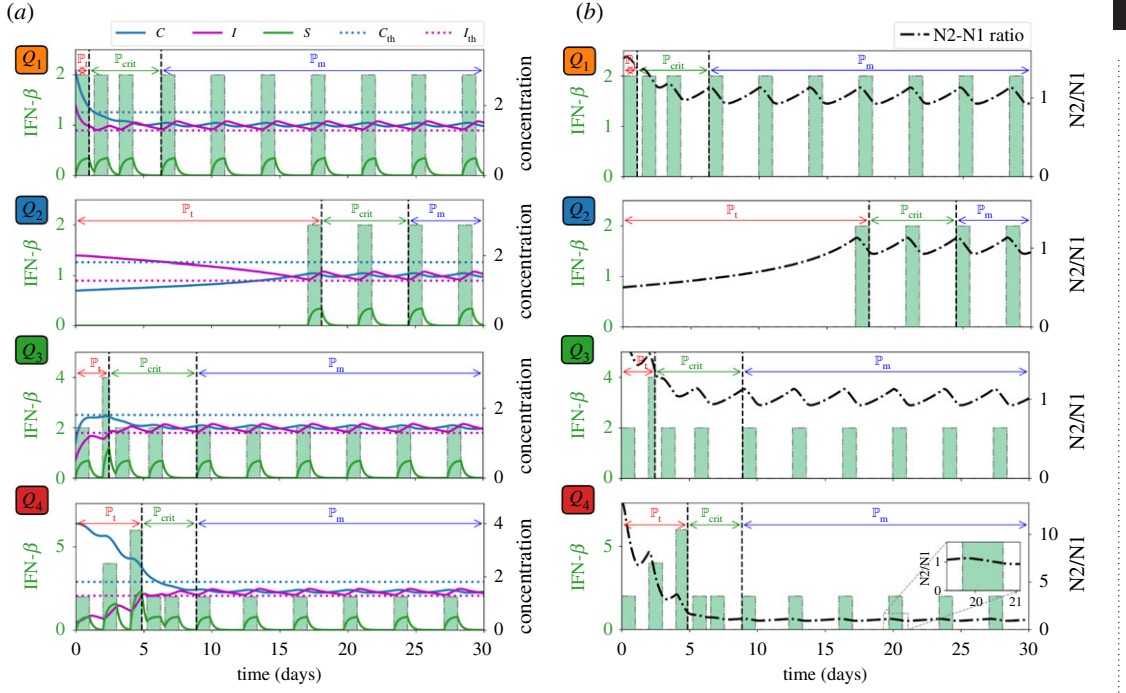

**Figure 5.** IFN-$\beta$ administration schedule under different initial conditions $Q_1$, $Q_2$, $Q_3$, $Q_4$ and corresponding dynamics of (a) $C$, $I$, $S$ and (b) N2/N1 levels. Administration of IFN-$\beta$ lowers the N2/N1 levels.

monitored and strategic IFN-$\beta$ infusion is necessary. Subsequently, $C$ and $I$ levels will be below and above their threshold values and IFN-$\beta$ can be regularly administered at minimum amount. This period is considered as the *maintenance period* $\mathbb{P}_m$.

Figure 5a illustrates the IFN-$\beta$ infusion control scheme, the corresponding IFN-$\beta$ concentration profiles, and dynamics of $C$ and $I$. It is shown that starting at $Q_4$ entails an increasing rate and amount of IFN-$\beta$ infusion, and longer $\mathbb{P}_t$. It is interesting to note that with initial condition at $Q_3$, the second infusion rate during $\mathbb{P}_t$ is 2 units but it can be verified that the amount of IFN-$\beta$ < 2. It can be observed in figure 5b that IFN-$\beta$ infusion decreases the N2/N1 ratio levels.

It is depicted in figure 6a that the total amount of IFN-$\beta$ needed for the entire medication duration for 60 days is the least when $(C, I)$ is initially at anti-tumorigenic region $Q_2$ as opposed to starting at tumorigenic region $Q_4$. In figure 6b, the dynamics of $C$–$I$ with different initial conditions under the IFN-$\beta$ control are steered towards the anti-tumorigenic region. Relatively efficient treatment results starting from anti-tumorigenic sector $Q_2$ and poor outcomes starting from tumorigenic status $Q_4$ are reflected in time courses of tumour volumes in figure 6c.

## 3.3. Concomitant TGF-$\beta$ inhibitor and IFN-$\beta$ controls

Now let us assume that both TGF-$\beta$ inhibitor and IFN-$\beta$ can be administered concurrently. As in the previous strategies, drug infusions can be done every other day where IFN-$\beta$ can be administered at an increasing amount. Both drugs are concomitantly administered until condition (3.1) is satisfied which spans the treatment period $\mathbb{P}_t$. After the infusions, $C$ and $I$ will decrease and increase, respectively, for some time. Before $C$ and $I$ cross their corresponding threshold value from below and above, respectively, simultaneous drug administration should be carried out. This duration covers the cancer-free period $\mathbb{P}_{cf}$. Then continuous monitoring of $C$ and $I$ profiles are necessary to determine the time for next concomitant administrations of TGF-$\beta$ inhibitor and IFN-$\beta$. These period is referred to as the relapse period $\mathbb{P}_r$. Concomitant infusion scheme, corresponding TGF-$\beta$ inhibitor and IFN-$\beta$ concentrations, and dynamics of the state variables starting at different initial conditions are depicted in figure 7a. The average levels of N2/N1 ratio for each regimen denoted by $\tau_{\mathbb{P}_t}$, $\tau_{\mathbb{P}_{cf}}$, $\tau_{\mathbb{P}_r}$ under various scenarios are depicted in figure 7b. Figure 8a shows a better comparison of the average amount of N1/N2 levels for each medication period under different conditions while figure 8b illustrates the ratio of the maximum ($\tau_{max}$) and minimum ($\tau_{min}$) N2/N1 levels for each condition. Obviously, $\tau_{max}/\tau_{min}$ is highest at high-risk state $Q_4$.

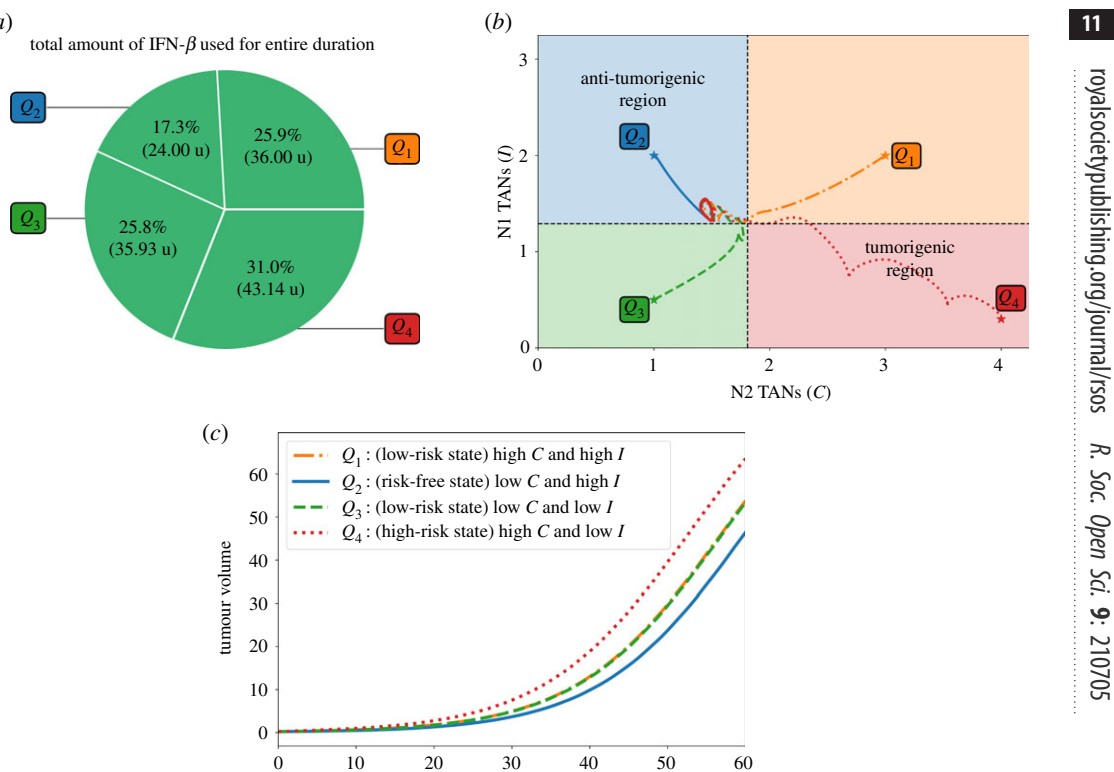

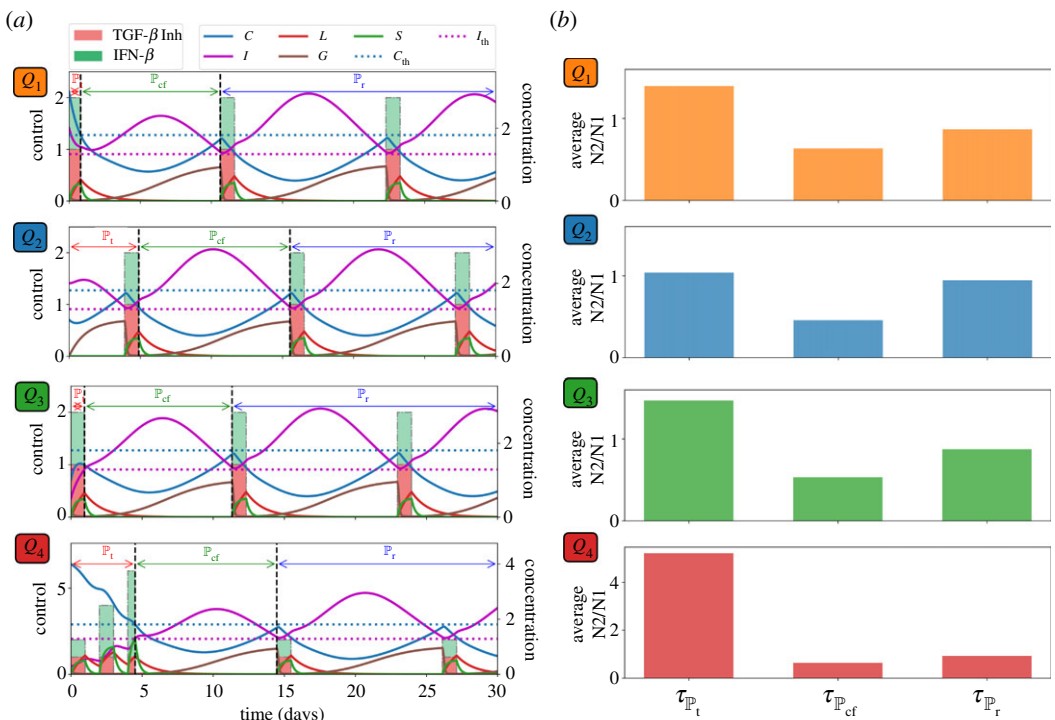

**Figure 6.** (*a*) Proportion of the amount of IFN-$\beta$ used for the entire treatment duration, (*b*) dynamics of the N1-N2 system and (*c*) tumour dynamics under IFN-$\beta$ control starting at different initial conditions $Q_1$, $Q_2$, $Q_3$, $Q_4$.

**Figure 7.** (*a*) Profiles of concomitant TGF-$\beta$ inhibitor and IFN-$\beta$ infusion protocol and corresponding concentration dynamics of $C$, $I$, $L$, $G$, $S$, and (*b*) average levels of N2/N1 ($\tau$) for each period ($\mathbb{P}_t$, $\mathbb{P}_{cf}$, $\mathbb{P}_r$) starting at different initial conditions $Q_1$, $Q_2$, $Q_3$, $Q_4$.

The total amount of TGF-$\beta$ inhibitor and IFN-$\beta$ drugs administered for the entire medication duration of 60 days are depicted in figure 9*a,b*, respectively. It can be observed that starting at $Q_4$ needs more infusions and thus more drug amount used. On the contrary, less administration and drug amount is needed when ($C$, $I$) is initially at risk-free state $Q_2$. Figure 9*c* shows the $C-I$ dynamics starting at different initial

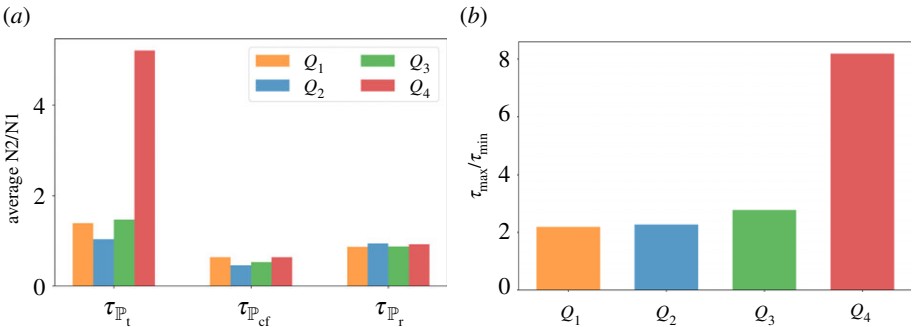

**Figure 8.** (a) Comparison of the average N2/N1 levels ($\tau$) for each medication period ($\mathbb{P}_t$, $\mathbb{P}_{cf}$, $\mathbb{P}_r$) and (b) ratio of the maximum ($\tau_{max}$) and minimum ($\tau_{min}$) N2/N1 levels for each condition $Q_1$, $Q_2$, $Q_3$, $Q_4$ under concomitant TGF-$\beta$ inhibitor and IFN-$\beta$ scheme.

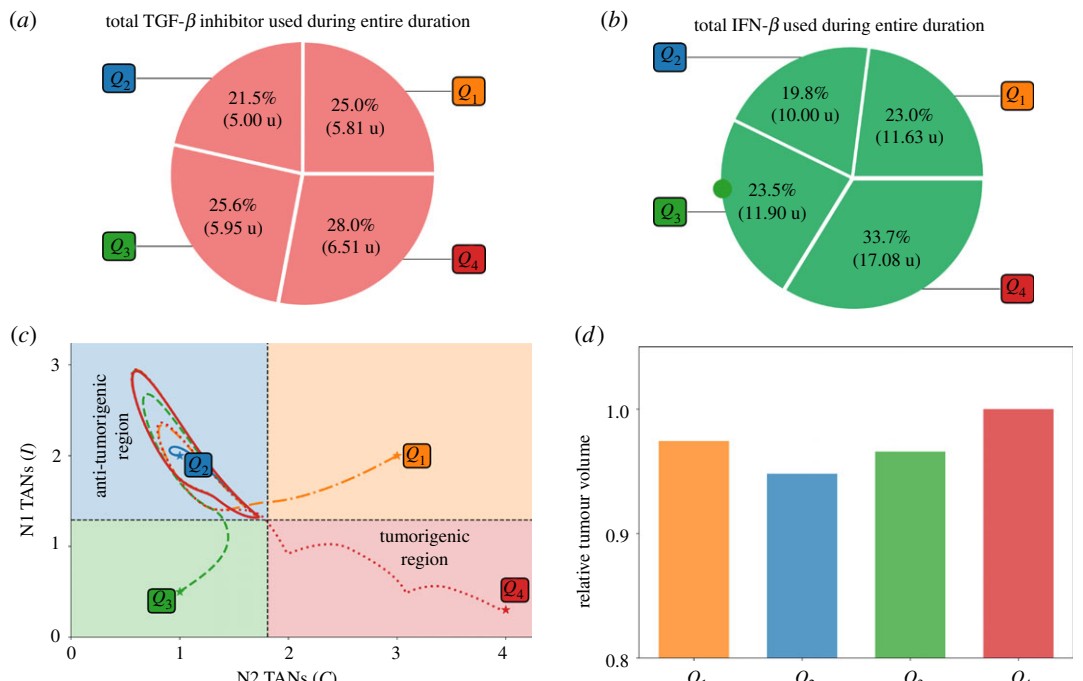

**Figure 9.** Proportion of the amount of TGF-$\beta$ inhibitor (a) and IFN-$\beta$ (b) used for the entire treatment duration, (c) dynamics of the N1-N2 system and (d) tumour volume relative to the high-risk state at the end simulation time under concomitant TGF-$\beta$ inhibitor and IFN-$\beta$ controls starting at different initial conditions $Q_1$, $Q_2$, $Q_3$, $Q_4$.

conditions. Administration of concomitant controls will drive all the trajectories to the anti-tumorigenic region. Scaled tumour volumes ($Q_4$ bar in figure 6d) and elongated path from the initial TAN distribution in the tumorigenic zone $Q_4$ (red dotted curve in figure 6c) also indicate the relatively worst outcome in decreasing the tumour size in the presence of a N2-dominant tumour microenvironment.

## 3.4. Alternating TGF-$\beta$ inhibitor and IFN-$\beta$ controls

Suppose that TGF-$\beta$ inhibitor and IFN-$\beta$ cannot be administered simultaneously due to toxicity, adverse drug reactions, etc. Here, we examine the scheme where drug infusion can be carried out alternately and every other day. Control scheme is obtained by analogously following the protocol and condition laid out in the previous strategies. The drug infusion protocol, its concentrations and state dynamics starting with different initial conditions are illustrated in figure 10a and the average levels of N2/N1 ratio $\tau_{\mathbb{P}_t}$, $\tau_{\mathbb{P}_{cf}}$, $\tau_{\mathbb{P}_r}$ under different conditions are illustrated in figure 10b. It is interesting to note that with initial condition at $Q_1$, IFN-$\beta$ infusion is not necessary a day after the first TGF-$\beta$ inhibitor administration. Also, starting at $Q_3$ needs lesser amount of IFN-$\beta$ for its first infusion. Figure 11a shows a comparison of the average amount of N1/N2 levels for each medication period under different conditions while figure 11b illustrates the ratio of the maximum ($\tau_{max}$) and minimum ($\tau_{min}$)

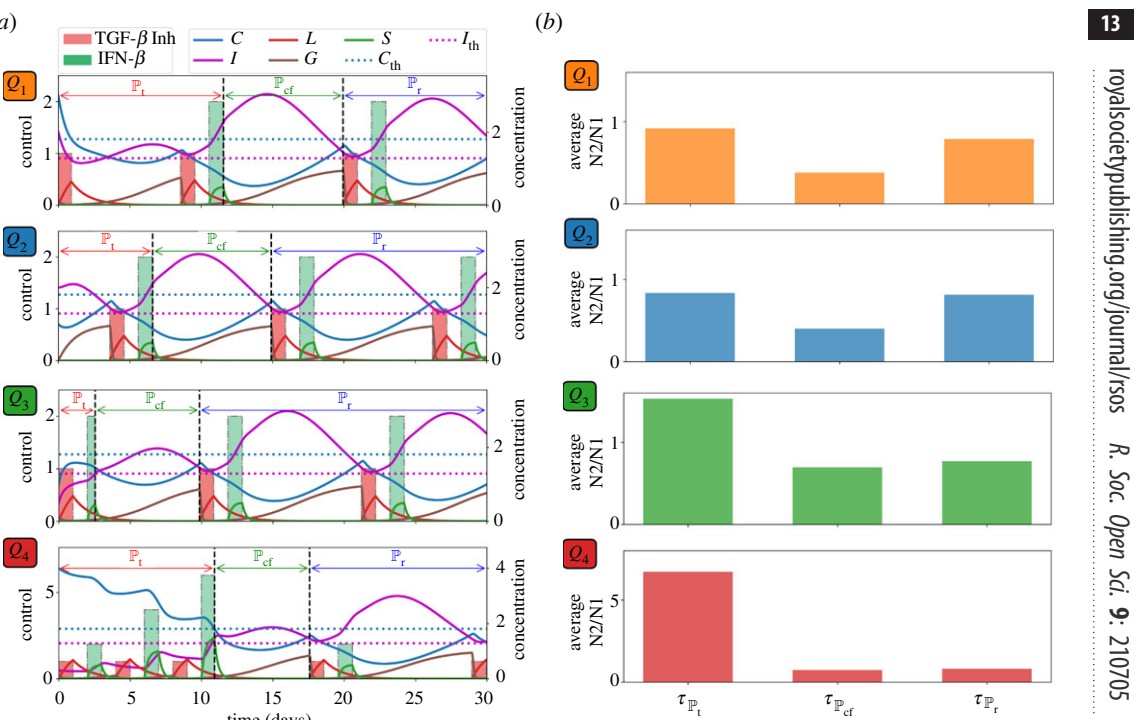

**Figure 10.** (*a*) Profiles of alternating TGF-$\beta$ inhibitor and IFN-$\beta$ infusion protocol and corresponding concentration dynamics of $C$, $I$, $L$, $G$, $S$, and (*b*) average levels of N2/N1 ($\tau$) for each treatment period ($\mathbb{P}_{t}$, $\mathbb{P}_{cf}$, $\mathbb{P}_{r}$) starting at different initial conditions $Q_1$, $Q_2$, $Q_3$, $Q_4$.

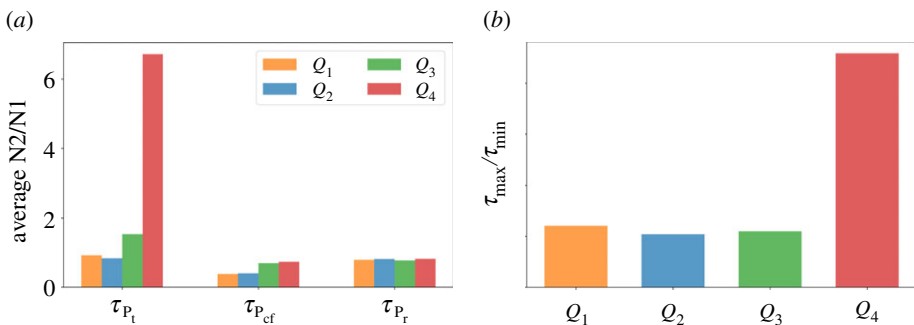

**Figure 11.** (*a*) Comparison of the average N2/N1 levels ($\tau$) for each medication period ($\mathbb{P}_{t}$, $\mathbb{P}_{cf}$, $\mathbb{P}_{r}$) and (*b*) ratio of the maximum ($\tau_{max}$) and minimum ($\tau_{min}$) N2/N1 levels for each condition $Q_1$, $Q_2$, $Q_3$, $Q_4$ under alternating TGF-$\beta$ inhibitor and IFN-$\beta$ scheme.

N2/N1 levels for each condition under alternating administration of TGF-$\beta$ inhibitor and IFN-$\beta$. As expected, $\tau_{max}/\tau_{min}$ is highest at high-risk state $Q_4$.

The proportion and the total amount of TGF-$\beta$ inhibitor and IFN-$\beta$ used for the entire duration of 60 days are depicted in figure 12*a,b*, respectively. As expected, starting at $Q_4$ needs more drug infusions and hence more drug amount compared with other initial conditions. Trajectories of the $C$−$I$ dynamics with different initial conditions towards the anti-tumorigenic region under the alternating control scheme are illustrated in figure 12*c*. Time courses of tumour volumes in four cases (figure 12*d*) show the worst outcome in decreasing the tumour size with the initial TAN distribution in the tumorigenic zone $Q_4$, indicating the critical role of the N1/N2 immune conditions. It is important to note that, in order to satisfy condition (3.1), the initial drug administration and infusion duration vary with different initial conditions. This in turn, influences the drug amount to be used within a specific period. Starting at $Q_2$, for instance, showed that drug infusion is not carried out during the first day. On the other hand, starting at $Q_4$ demands several drug administrations to meet the condition (3.1).

Figure 13 compares the relative total amount of TGF-$\beta$ inhibitor and IFN-$\beta$ used in the proposed therapeutic strategies starting at different initial conditions. Across the various schemes, starting at $Q_4$ needs a bigger amount to bring $C$ below and keep $I$ above their respective threshold values.

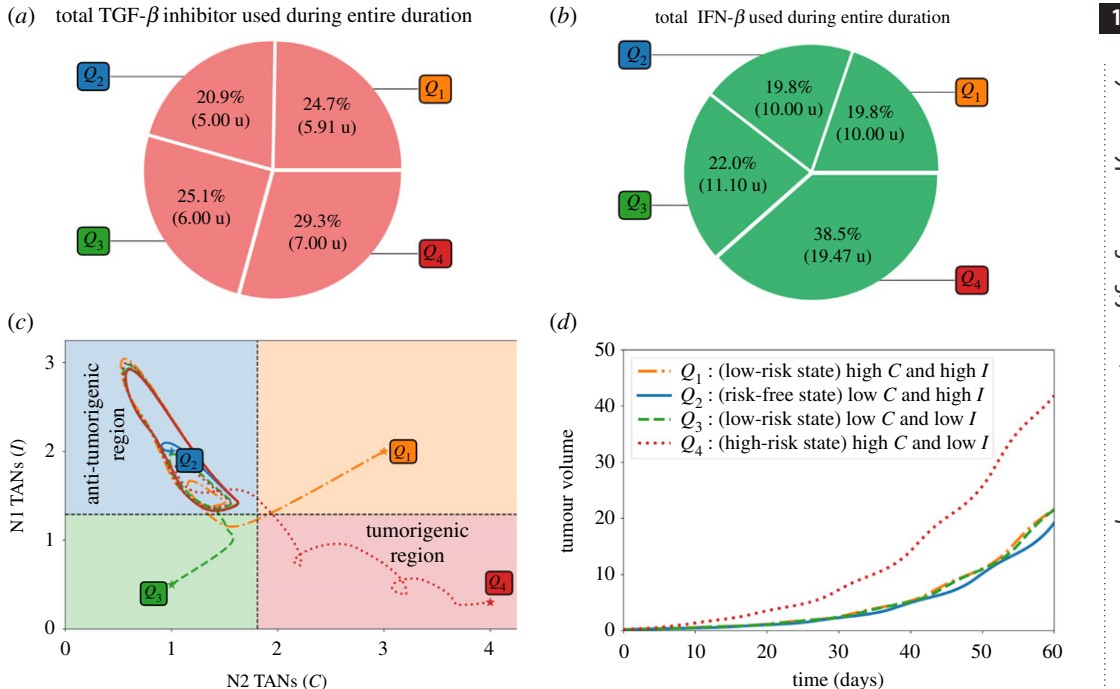

**Figure 12.** (*a*) Proportion of the amount of TGF-$\beta$ inhibitor, (*b*) IFN-$\beta$ used for the entire treatment duration, (*c*) dynamics of the N1-N2 system and (*d*) tumour dynamics under alternating TGF-$\beta$ inhibitor and IFN-$\beta$ controls starting at different initial conditions $Q_1$, $Q_2$, $Q_3$, $Q_4$.

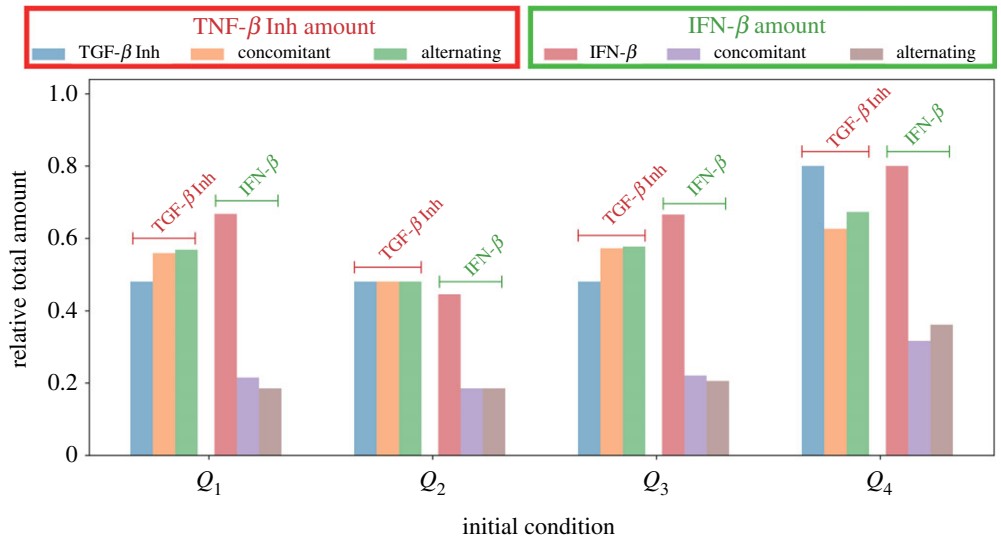

**Figure 13.** Relative amount of TGF-$\beta$ inhibitor and IFN-$\beta$ used among different control strategies.

## 4. Conclusion

The exact role of immune cells such as NK cells [4] and TANs in tumour growth is still poorly understood. Several studies have reported that neutrophils are cytotoxic to tumour cells. On the contrary, a growing clinical evidence supports that neutrophils can promote tumour progression by enhancing proliferation and angiogenesis, and inducing cell migration and metastasis [20,21,23]. Hence, TANs can display two different phenotypes: an anti-tumour (N1) and pro-tumour (N2) effector cells [15,16,24]. The current study investigates a regulatory model between N1 and N2 TANs. In this work, optimal control theory is employed to determine the amount, infusion times and cost of TGF-$\beta$ inhibitor and IFN-$\beta$ administrations to activate N1 leading to tumour suppression. This framework has been utilized to identify treatment schedules and anti-invasion therapy for

glioblastoma [90,91]. Both single and combination drug therapies are considered under various phenotypical conditions of N1 and N2. An important aspect of determining treatment protocol is the patients' pathophysiological condition. We identified therapeutic regimen for promoting anti-tumoural TANs and suppressing pro-tumoural neutrophil phenotypic states through four different administration of the TGF-$\beta$ inhibitor and IFN-$\beta$ in lung cancer. The optimal control scheme predicts that, depending on relative states of N1 and N2 phenotypes, therapy schedules may have to be adjusted properly to minimize adverse effects of these drugs and its administration cost in addition to maximizing anti-tumour efficacy.

Proper diagnosis of N1 and N2 activities are important tools in order to have an adaptive and individualized drug infusion protocol [60]. Converting N2 TANs to N1 could provide new therapeutic options for cancer treatment. Our work essentially can provide a general framework of *optimally controlled* injection schedules of anti-cancer drugs such as TGF-$\beta$ inhibitors and IFN-$\beta$ through immune control, i.e. a critical transition between N1 and N2 TANs. Immune cells in a tumour microenvironment play such a critical role in regulation of tumour growth and anti-cancer therapies [4], having two faces of anti-tumour and tumour-promoting roles. The mathematical framework in this work may be adapted for optimal results of anti-cancer therapies as far as appropriate patient-specific data in clinics are provided.

Our study has four limitations:

— The current work does not consider the role of other factors in TME such as intracellular pathways such as STAT1/STAT3/JAK [92] in response to IFN-$\beta$, TGF-$\beta$, and other stimuli [93], neutrophil elastase (NE) in the presence [94] and absence [95,96] of LPS, tumour-promoting or -suppressive immune response of NK cells [4,97], tumour-associated fibroblasts (TAFs) [67,98–100], ECM remodelling [69,101–103], angiogenesis from blood vessels [104–106], or growth factors such as EGF [107,108] and CSF-1 [5,109]. NK cells were shown to mediate dual roles of neutrophils in metastatic colonization [110,111]. For example, neutrophils can enhance extravasation of circulating tumour cells by suppressing NK cells [111]. Depletion or adjuvant therapy of NK cells was shown to increase anti-tumour efficacy in a combination therapy (OV-bortezomib-NK) relative to control, showing nonlinear behaviour of immune system [4]. These factors may play major roles in regulation of N1 → N2 transition, thus cancer progression. In particular, we focused on the effect of TGF-$\beta$ inhibitor and IFN-$\beta$ on TANs in this study, but it would be important to see the effect of those two inhibitors on NK cells and TAFs in future work. For example, TGF-$\beta$ and its inhibitor [98,99,112] and IFNs [112–115] play an important role in regulation of tumour-associated fibroblasts in cancer progression. We plan to include those players in a future optimal-control model.

— NET and NE were shown to promote tumour growth and invasion [116] by turning on the multiple signalling pathways including PI3K in lung cancer cells [117]. For example, upregulated NET activities near tumour sections induce the transformation of B cells [118], contributing to cancer progression [119]. In particular, mathematical models [116] and experimental data [120] suggest that NET can mediate the critical metastatic process [121,122] to stabilize the circulating tumour cells in the bloodstream and help extravasation of these cancerous cells. However, NET was also suggested to suppress tumour growth in colonic adenocarcinoma [123]. In our study, we did not take into account these critical influence of NETs in a spatial domain. We plan to develop an optimal control approach in a new framework of the partial differential equations of NETs in order to improve the therapeutic, anti-invasion strategies.

— Unforeseen microenvironmental factors may limit bi-lateral switches between N1 and N2 TANs. In our study, optimal control was applied only to IFN-$\beta$ and TGF-$\beta$ inhibitors. We plan to investigate the specific role and optimization of these stimuli in TME for better understanding of the role of TANs (either promotion [124–127] or suppression of tumour progression [123]). A new optimal control method has to be developed for a possible triple combination therapy, i.e. TGF-$\beta$ + TGF-$\beta$ inhibitor + immune agents [128]. We plan to develop an optimal control of the TAN's plasticity of the possibly continuous spectrum of the N1 → N2 switches, which requires better understanding and experiments of the biological system as well as advanced optimal control theory.

— It has been reported that a more realistic clinical situation employs linear cost functions. These types of problems have been used in several works devoted to the administration of single and combination therapies to treat different types of tumours [129–133]. On the contrary, a study by Glick & Mastroberardino [74] concluded that quadratic control yields continuous, low doses of the therapeutic drug producing a better outcome in the eradication of the solid tumour. Several researches on optimal control approaches for cancer treatment still favour quadratic controls [77–79].

Under certain circumstances, both linear and quadratic controls obtain qualitatively similar results [76,81,134]. It is true that profiles of treatment strategies depend on the landscape of the cost functionals. These differences show the importance of carefully defining an objective functional that most accurately reflects the toxicities of a particular drug along with the objective of the treatment strategy, for instance, decreasing the tumour mass at the end of the treatment interval, reducing the overall tumour burden over the treatment interval, and some other clinically relevant criteria [73]. It is therefore suggested that further model iterations should include exhaustive investigations on linear controls to have holistic treatment strategies for cancer treatment. Further analysis of models with other types of objective functionals should also be pursued. Varying results should then be presented to the medical practitioner for them to decide which solutions best fit the biological situation and can be used for practical implementation, if any [134].

This work, however, provides a general framework of optimal control approach for the fundamental transition from N2 to N1 TANs, thus inhibiting tumour growth, in response to known key players.

Typical lifespan of neutrophils is usually under tight control for maintenance of tissue homeostasis due to their potential toxicity [135], leading to short half-life in blood circulation after leaving the bone marrow. However, there exist multiple cellular and molecular factors such as smoking and reactive oxygen species (ROS) [136–138] that can influence their longevity [139,140], altered signalling pathways and neutrophil apoptosis in TME [136,137,141,142]. For example, ROS production is shown to be decreased in the absence of endogenous IFN-$\beta$, inducing cellular delays in apoptosis of TANs [143]. Since these factors may affect a complex imbalance between N1 and N2 TANs, an optimal control framework of such N1/N2 system with *time delays* [144,145] is needed in order to control the aggressive promotion of aggressive tumour growth.

Building a mathematical model does not entail all important details of a life process, rather, distil key elements that could provide insight into the underlying mechanisms and generate novel hypotheses for experimentation [45]. Since no model is perfect, and is lacking some aspects of reality, a symbiotic approach incorporating *in vivo, in vitro* and *in silico* techniques could be proven to be beneficial [146,147]. Despite several caveats and limitations, mathematical modelling will still be instrumental in understanding cancer biology and treatment [47]. Further design of experiments and development of computational techniques for verification and validation should be carried out using experimental data. Various types of mathematical models were suggested for better understanding of tumour biology: (i) ordinary differential equations [60,91,148–152], (ii) delay differential equations [60,148,153], (iii) partial differential equations [4,5,67,68,101,103,116,154–160], (iv) immersed boundary method [161,162], (v) individual-based models [163–165], and (vi) multi-scale hybrid model [69,100,166–173]. See [174] for a review of cell-based approaches. These mathematical approaches have been applied to investigation of various aspects of signalling network of oncogenes and tumour suppressors, tumour growth, cellular invasion, angiogenesis, recurrence, interaction with tumour microenvironment including chemokines/cytokines, stromal cells, immune cells, metastasis and development of anti-cancer strategies. In particular, multi-scale mathematical models [69,100,166,167,169–173,175,176] could be used to address dynamical changes in inter- and intra-cellular signalling pathways at the microscale level and integrate cellular process at the cellular level. We hope to address these issues in future work.

Data accessibility. We deposited all codes and data within the Dryad Digital Repository: https://doi.org/10.5061/dryad.g79cnp5r0. We have two files: (1) README.txt (2) data_RSOS-210705.zip. Here, the zip file contains all codes and data.

Competing interests. The authors declare that they have no competing interests.

Funding. This work was supported by the National Research Foundation of Korea (NRF) grant funded by the Korea government (MSIT) (No. 2021R1A2C1010891) (Y.K.).

Acknowledgements. A.D.L.R.V. acknowledges the support of the Institute of Mathematics, University of the Philippines Diliman and the Institute for Basic Science (IBS-R029-C3).

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
