## [Peer Review File · Royal Society Open Science]

Review History

RSOS-210705.R0 (Original submission)

Review form: Reviewer 1

Is the manuscript scientifically sound in its present form?

Yes

Are the interpretations and conclusions justified by the results?

Yes

Is the language acceptable?

Yes

Do you have any ethical concerns with this paper?

No

Have you any concerns about statistical analyses in this paper?

No

Recommendation?

Accept with minor revision (please list in comments)

Comments to the Author(s)

Dear Editor,

The article presents a novel model of antitumor and protumor neutrophils, which they call N1 and N2 neutrophils, respectively. Then they formulate an optimal control problem in which they administer cytokine therapy to shift the balance of neutrophils towards the antitumor N1 state to help treat the tumor.

I found the work interesting and well organized. They present the results of four optimal control scenarios systematically and thoroughly. It is a firm basis for people, including the authors themselves, to build on to further study this area.

Minor comments:

In Figure 1, the authors need to define L and S earlier, so that Figure 1 makes sense. They cannot only define L and S long after presenting Figure 1. In addition to defining these variables in the text, they ought to also define L and S (and G, C, I) in the figure caption for easy referencing.

On p. 4 at model (1), they authors should describe the model thoroughly and explain the terms even if the model already explained in reference [54]. The reader should not be required to look up [54] to understand the model.

In Figure 2 and elsewhere, make sure all the fonts in the figures are at least 10 point.

On p. 6, in Equations (5) to (8), why did the authors pick these functions to optimize? The authors ought to include an explanation of the reasons?

Review form: Reviewer 2

Is the manuscript scientifically sound in its present form?

No

Are the interpretations and conclusions justified by the results?

No

Is the language acceptable?

Yes

Do you have any ethical concerns with this paper?

No

Have you any concerns about statistical analyses in this paper?

No

Recommendation?

Major revision is needed (please make suggestions in comments)

Comments to the Author(s)

In this manuscript entitled "Optimal regulation of tumor-associated neutrophils in cancer progression," based on the report that TAN (tumor-associated neutrophils) status in the tumor microenvironment regulates tumor growth, the authors developed a mathematical model that TGF- β inhibitor and/or IFN- β administration promotes N2 to N1 polarization of TAN and suppresses tumor growth. Then, the authors derived optimal therapeutic strategy for cancer using TGF- β inhibitor and/or IFN- β administration using optimal control theory.

The manuscript demonstrates a large amount of simulation data but seems to lack a clear interpretation of the findings of this study. The authors should clearly show the significance and limitation of this study. Listed below are my specific comments.

1. This manuscript, especially in the Introduction and Conclusion sections, covers a wide range of topics with many references. As a result, this reviewer could not clearly understand the conclusions of this study and thinks this paper should be more concise and focused only on the main point.

2. Since TGF- β inhibitor and IFN- β affect a broader range of cell types in the tumor microenvironment other than the TAN, Treg, CD8+, and Th17 cells that considered in the mathematical model, the simulations in this study may be focused on the limited effects of TGF- β inhibitor and IFN- β in the regulation of tumor growth. Of course, it is difficult to take all factors into account in a mathematical model. Still, the limitation should be clarified, for example, that this mathematical model ignores the effects of TGF- β inhibitor and IFN- β on NK cells, tumor-associated fibroblasts, and so on.

3. The half-life of circulating neutrophils is reported to be 19 hours (Lahoz-Beneytez et al., Blood, 2016). Therefore, this reviewer speculates that all TANs would be replaced in a few days, although the tumor microenvironment may prolong the life of TANs as the authors described in the Discussion.

In Figure 5, for example, the N2/N1 ratio is kept around 1.0 under the conditions that IFN- β remains at zero concentration for several days after degradation. However, this reviewer concerns that the N2/N1 ratio would return to the initial condition during the IFN- β -free period due to the turnover of TANs.

4. The differential equations (2) consist of 6 variables (L, G, C, S, I, and T). If this reviewer understands correctly, the time-course of L, G, C, S, and I are shown according to the four optimal control scenarios (Figures 3, 5, 7, and 10), but "T" does not appear in any figures. Therefore, the authors should show the time-course of T, how the tumor size is regulated in response to the treatment of TGF- β inhibitor and/or IFN- β .

I have a few additional minor comments, explained below.

5. In the third paragraph of the Introduction, the authors say, "TGF- β inhibitor such as gefitinib." However, gefitinib is an EGFR inhibitor, not a TGF- β inhibitor.

6. The authors assumed that TGF- β inhibitor and IFN- β can be administrated every other day in the simulations, but what is the evidence?

Review form: Reviewer 3

Is the manuscript scientifically sound in its present form?

Yes

Are the interpretations and conclusions justified by the results?

Yes

Is the language acceptable?

Yes

Do you have any ethical concerns with this paper?

No

Have you any concerns about statistical analyses in this paper?

No

Recommendation?

Accept with minor revision (please list in comments)

Comments to the Author(s)

In the paper the authors consider a mathematical model of tumor-associated neutrophils in lung cancer progression. This model uses a mathematical model to explain the optimal control problems of TGF β and INF β . The part of the mathematical model has been revised in detail in a previous review of another Journal. However, there are some questions about the contents of the paper. Below are these comments:

1, The authors define Q1 to Q4 in the Results and Discussion section, but how do you apply them to individual cases in the actual clinical settings? How do you measure the level of N1 and N2 component? Is it by a blood test or histopathological examination such as surgical specimens or biopsy specimens?

2, Have you verified the results of this mathematical model in actual clinical cases or in vitro? If it can be shown that the results of actual biological research and mathematical models match, I think it will be a more meaningful paper in the clinical settings. However, it does not force the authors to do biological research when it is difficult to research.

Decision letter (RSOS-210705.R0)

Dear Dr Kim

The Editors assigned to your paper RSOS-210705 "Optimal regulation of tumor-associated neutrophils in cancer progression" have now received comments from reviewers and would like you to revise the paper in accordance with the reviewer comments and any comments from the Editors. Please note this decision does not guarantee eventual acceptance.

Please submit your revised manuscript and required files (see below) no later than 21 days from today's (ie 11-Aug-2021) date. Note: the ScholarOne system will 'lock' if submission of the revision is attempted 21 or more days after the deadline. If you do not think you will be able to meet this deadline please contact the editorial office immediately.

on behalf of Professor Takashi Suzuki (Associate Editor) and Mark Chaplain (Subject Editor)
openscience@royalsociety.org

Associate Editor Comments to Author (Professor Takashi Suzuki):

Although the referees are basically positive for publications of your paper, some more efforts are necessary. Please take regards their comments to improve it.

Reviewer comments to Author:

Reviewer: 1
Comments to the Author(s)
Dear Editor,

The article presents a novel model of antitumor and protumor neutrophils, which they call N1 and N2 neutrophils, respectively. Then they formulate an optimal control problem in which they administer cytokine therapy to shift the balance of neutrophils towards the antitumor N1 state to help treat the tumor.

I found the work interesting and well organized. They present the results of four optimal control scenarios systematically and thoroughly. It is a firm basis for people, including the authors themselves, to build on to further study this area.

Minor comments:

In Figure 1, the authors need to define L and S earlier, so that Figure 1 makes sense. They cannot only define L and S long after presenting Figure 1. In addition to defining these variables in the text, they ought to also define L and S (and G, C, I) in the figure caption for easy referencing.

On p. 4 at model (1), they authors should describe the model thoroughly and explain the terms even if the model already explained in reference [54]. The reader should not be required to look up [54] to understand the model.

In Figure 2 and elsewhere, make sure all the fonts in the figures are at least 10 point.

On p. 6, in Equations (5) to (8), why did the authors pick these functions to optimize? The authors ought to include an explanation of the reasons?

Reviewer: 2

Comments to the Author(s)

In this manuscript entitled "Optimal regulation of tumor-associated neutrophils in cancer progression," based on the report that TAN (tumor-associated neutrophils) status in the tumor microenvironment regulates tumor growth, the authors developed a mathematical model that TGF- β inhibitor and/or IFN- β administration promotes N2 to N1 polarization of TAN and suppresses tumor growth. Then, the authors derived optimal therapeutic strategy for cancer using TGF- β inhibitor and/or IFN- β administration using optimal control theory.

The manuscript demonstrates a large amount of simulation data but seems to lack a clear interpretation of the findings of this study. The authors should clearly show the significance and limitation of this study. Listed below are my specific comments.

1. This manuscript, especially in the Introduction and Conclusion sections, covers a wide range of topics with many references. As a result, this reviewer could not clearly understand the conclusions of this study and thinks this paper should be more concise and focused only on the main point.

2. Since TGF- β inhibitor and IFN- β affect a broader range of cell types in the tumor microenvironment other than the TAN, Treg, CD8+, and Th17 cells that considered in the mathematical model, the simulations in this study may be focused on the limited effects of TGF- β inhibitor and IFN- β in the regulation of tumor growth. Of course, it is difficult to take all factors into account in a mathematical model. Still, the limitation should be clarified, for example, that this mathematical model ignores the effects of TGF- β inhibitor and IFN- β on NK cells, tumor-associated fibroblasts, and so on.

3. The half-life of circulating neutrophils is reported to be 19 hours (Lahoz-Beneytez et al., Blood, 2016). Therefore, this reviewer speculates that all TANs would be replaced in a few days, although the tumor microenvironment may prolong the life of TANs as the authors described in the Discussion.

In Figure 5, for example, the N2/N1 ratio is kept around 1.0 under the conditions that IFN- β remains at zero concentration for several days after degradation. However, this reviewer concerns that the N2/N1 ratio would return to the initial condition during the IFN- β -free period due to the turnover of TANs.

4. The differential equations (2) consist of 6 variables (L, G, C, S, I, and T). If this reviewer understands correctly, the time-course of L, G, C, S, and I are shown according to the four optimal control scenarios (Figures 3, 5, 7, and 10), but "T" does not appear in any figures.

Therefore, the authors should show the time-course of T, how the tumor size is regulated in response to the treatment of TGF- β inhibitor and/or IFN- β .

I have a few additional minor comments, explained below.

5. In the third paragraph of the Introduction, the authors say, "TGF- β inhibitor such as gefitinib." However, gefitinib is an EGFR inhibitor, not a TGF- β inhibitor.

6. The authors assumed that TGF- β inhibitor and IFN- β can be administrated every other day in the simulations, but what is the evidence?

Reviewer: 3

Comments to the Author(s)

In the paper the authors consider a mathematical model of tumor-associated neutrophils in lung cancer progression. This model uses a mathematical model to explain the optimal control problems of TGF β and INF β . The part of the mathematical model has been revised in detail in a previous review of another Journal. However, there are some questions about the contents of the paper. Below are these comments:

1, The authors define Q1 to Q4 in the Results and Discussion section, but how do you apply them to individual cases in the actual clinical settings? How do you measure the level of N1 and N2 component? Is it by a blood test or histopathological examination such as surgical specimens or biopsy specimens?

2, Have you verified the results of this mathematical model in actual clinical cases or in vitro? If it can be shown that the results of actual biological research and mathematical models match, I think it will be a more meaningful paper in the clinical settings. However, it does not force the authors to do biological research when it is difficult to research.

===PREPARING YOUR MANUSCRIPT===

===PREPARING YOUR REVISION IN SCHOLARONE===

Author's Response to Decision Letter for (RSOS-210705.R0)

See Appendix A.

RSOS-210705.R1 (Revision)

Review form: Reviewer 1

Is the manuscript scientifically sound in its present form?

Yes

Are the interpretations and conclusions justified by the results?

Yes

Is the language acceptable?

Yes

Do you have any ethical concerns with this paper?

No

Have you any concerns about statistical analyses in this paper?

No

Recommendation?

Accept as is

Comments to the Author(s)

Dear Editor,

The authors have satisfactorily addressed my comments.

Some minor comments are

In p. 4 under Equation (1), should say something like "in the equation for C" rather than "in Eq of 'C'" and do the same for I, T, and N.

Also, the authors should go through the manuscript and fix up the English phrasing and grammar.

Review form: Reviewer 2

Is the manuscript scientifically sound in its present form?

Yes

Are the interpretations and conclusions justified by the results?

Yes

Is the language acceptable?

Yes

Do you have any ethical concerns with this paper?

No

Have you any concerns about statistical analyses in this paper?

No

Recommendation?

Accept as is

Comments to the Author(s)

The authors have responded to all the concerns raised by this reviewer.

Decision letter (RSOS-210705.R1)

Dear Dr Kim,

It is a pleasure to accept your manuscript entitled "Optimal regulation of tumor-associated neutrophils in cancer progression" in its current form for publication in Royal Society Open Science. The comments of the reviewer(s) who reviewed your manuscript are included at the foot of this letter.

on behalf of Professor Takashi Suzuki (Associate Editor) and Mark Chaplain (Subject Editor)
openscience@royalsociety.org

Reviewer comments to Author:
Reviewer: 1
Comments to the Author(s)
Dear Editor,

The authors have satisfactorily addressed my comments.

Some minor comments are

In p. 4 under Equation (1), should say something like "in the equation for C" rather than "in Eq of 'C'" and do the same for I, T, and N.

Also, the authors should go through the manuscript and fix up the English phrasing and grammar.

Reviewer: 2
Comments to the Author(s)
The authors have responded to all the concerns raised by this reviewer.

Appendix A

Optimal regulation of tumor-associated neutrophils in lung cancer progression

Aurelio A. de los Reyes V and Yangjin Kim

RESPONSE TO REVIEWERS' COMMENTS

October 23, 2021

REFEREE 1:

The article presents a novel model of antitumor and protumor neutrophils, which they call N1 and N2 neutrophils, respectively. Then they formulate an optimal control problem in which they administer cytokine therapy to shift the balance of neutrophils towards the antitumor N1 state to help treat the tumor.

I found the work interesting and well organized. They present the results of four optimal control scenarios systematically and thoroughly. It is a firm basis for people, including the authors themselves, to build on to further study this area.

Minor comments:

- In Figure 1, the authors need to define L and S earlier, so that Figure 1 makes sense. They cannot only define L and S long after presenting Figure 1. In addition to defining these variables in the text, they ought to also define L and S (and G, C, I) in the figure caption for easy referencing.

(Response) Thank you for careful reading and suggestions. We now defined the main variables at the end of Introduction Section so that it can naturally appear before Fig 1. So, we added the following phrase :

“In this work, we consider the following variables in a mathematical model:

$C(t)$ = density of the N2 complex at time t ;
 $I(t)$ = density of the N1 complex at time t ;
 $T(t)$ = tumor volume at time t ;
 $G(t)$ = concentration of TGF- β at time t ;
 $L(t)$ = concentration of TGF- β inhibitor at time t ;
 $S(t)$ = concentration of IFN- β at time t ;"

[page 2, 5th paragraph, lines 1-7, in the (marked) revised manuscript]

In addition, in order to make it consistent in the main text, we removed the definition of L, S, G by replacing

“dimensionless form where TGF- β inhibitor and TGF- β are denoted by L and G , respectively; N1 and N2 complexes are grouped as module C and I , respectively; and IFN- β is represented by S .”

with

“dimensionless form; N1 and N2 complexes are grouped as module C and I , respectively.”

[page 6, 1st paragraph, line 7, in the (marked) revised manuscript]

- On p.4 at model (1), they authors should describe the model thoroughly and explain the terms even if the model already explained in reference [54]. The reader should not be required to look up [54] to understand the model.

(Response) Thank you for careful reading and suggestion. In order to assist the understanding of the model without looking up Kim et al, 2019 [28] as follows:

$$\begin{aligned}
 \frac{dC}{dt} &= \underbrace{\lambda}_{\text{source (IL-6)}} + \underbrace{\lambda_G G}_{\text{source (TGF-}\beta\text{)}} + \underbrace{\frac{k_1}{k_3^2 + \alpha I^2}}_{\text{inhibition from N1}} - \underbrace{C}_{\text{decay}}, \\
 \frac{dI}{dt} &= \underbrace{\lambda_S S}_{\text{source (IFN-}\beta\text{)}} + \underbrace{\frac{k_2}{k_4^2 + \beta C^2}}_{\text{inhibition from N2}} - \underbrace{\mu I}_{\text{decay}}, \\
 \frac{dT}{dt} &= \underbrace{r \left(1 + \frac{C}{K + \gamma_1 I}\right) T \left(1 - \frac{T}{T_0}\right)}_{\text{growth}} - \underbrace{\delta IT}_{\text{killing}},
 \end{aligned} \tag{1}$$

[page 4, Eq (1), in the (marked) revised manuscript]

In addition, we added the following statements below the equations above

in order to explain the model:

“Here, two sources (IL-6, TGF- β) of the N2 complex are represented in the first and second terms in Eq of ‘C’ while the IFN- β -mediated source of the N1 complex is provided in the first term in Eq of ‘I’. The third term in Eq of ‘C’ and second term in Eq of ‘I’ represent the mutual inhibition between N1 and N2, respectively. Tumor growth and N1-mediated tumor cell killing are represented in the first and second term in Eq of ‘T’, respectively. Finally, decay process of the N1 and N2 complexes in the last terms in Eq of ‘C’ and ‘N’, respectively. ”

[page 4, lines 1-6 starting from Eq (1), in the (marked) revised manuscript]

- In Figure 2 and elsewhere, make sure all the fonts in the figures are at least 10 point.

(Response) Thank you for careful reading. We increased all the fonts in all figures (from Fig 1 to Fig 13). Now, those figures look much better with the larger fonts.

In addition, we made the following changes (blue) for caption of Fig 2 by replacing

“Nullclines of model (1) in the $C - I$ phase plane in response to (A) low, $G = 0.1$, (B) intermediate, $G = 0.4$, and (C) high, $G = 1$ levels of TGF- β showing the corresponding steady states where $SS^{(s)}$ and $SS^{(u)}$ denote stable, respectively, unstable steady state. Hysteresis diagram with respect to varying signals of (D) TGF- β (G) and (E) IFN- β (S) promoting N1-N2 on-off switch activation. (E) Codimension 2 bifurcation for different G and S levels depicting division of bistable and monostable region and a cusp point (CP).”

with

“(A-C) Nullclines of model (1) in the $C - I$ phase plane in response to (A) low, $G = 0.1$, (B) intermediate, $G = 0.4$, and (C) high, $G = 1$ levels of TGF- β showing the corresponding steady states where $SS^{(s)}$ and $SS^{(u)}$ denote stable, respectively, unstable steady state. ***Anti-tumorigenic and tumorigenic regions are marked in blue and pink boxes in (A-C). (D-E) Hysteresis diagram with respect to varying signals of TGF- β and IFN- β promoting N1-N2 on-off switch activation. (F) Codimension 2 bifurcation for different G and S levels depicting division of bistable and monostable region and a cusp point (CP). ”**

[page 5, caption of Fig 2, in the (marked) revised manuscript]

- On p.6, in Equations (5) to (8), why did the authors pick these functions to optimize? The authors ought to include an explanation of the reasons?

(Response) Thank you for careful suggestions. Some of reasons are already mentioned in the text. For instance, in equation 5, it is written: “TGF- β

inhibitor control only. In this scheme, we want to investigate the anti-tumor effect of TGF- β inhibitor on tumor growth.” In addition, following the reviewer’s suggestion, we now made the following changes:

We replaced “The OCP” with “In particular, we want to minimize the tumor size (T) and dose of TGF- β inhibitor while keeping the same IFN- β supply. Thus, the OCP”

[page 7, 1st paragraph, lines 2-3, in the (marked) revised manuscript]

We replaced “The problem” with “In particular, we want to minimize the tumor size (T) and dose of IFN- β in the absence of IFN- β inhibitor. Thus, the problem” [page 7, 2nd paragraph, lines 1-3, in the (marked) revised manuscript]

We replaced “The OCP ” with “Specifically, we want to minimize the tumor size (T) and doses of both TGF- β inhibitor and IFN- β . ”

[page 7, 3rd paragraph, lines 2-3, in the (marked) revised manuscript]

We replaced “The OCP ” with “Specifically, we want to minimize the tumor size (T) and doses of both TGF- β inhibitor and IFN- β in an alternating injection scheme. Thus, the OCP”

[page 7, 4th paragraph, lines 2-4, in the (marked) revised manuscript]

In addition to these changes, we made the following minor changes in the (marked) revised manuscript in order to meet the journal requirement (from editorial staffs)

(i) “Ohio, ” \rightarrow “OH 43210, ” [page 1, author affiliation]

REFeree 2:

In this manuscript entitled ”Optimal regulation of tumor-associated neutrophils in cancer progression,” based on the report that TAN (tumor-associated neutrophils) status in the tumor microenvironment regulates tumor growth, the authors developed a mathematical model that TGF- β inhibitor and/or IFN- β administration promotes N2 to N1 polarization of TAN and suppresses tumor growth. Then, the authors derived optimal therapeutic strategy for cancer using TGF- β inhibitor and/or IFN- β administration using optimal control theory.

The manuscript demonstrates a large amount of simulation data but seems to lack a clear interpretation of the findings of this study. The authors should clearly show the significance and limitation of this study. Listed below are my specific comments.

1. This manuscript, especially in the Introduction and Conclusion sections, covers a wide range of topics with many references. As a result, this reviewer could not clearly un-

derstand the conclusions of this study and thinks this paper should be more concise and focused only on the main point.

(Response) In order to make it more structured and focused, we now added the following paragraph in the INTRODUCTION SECTION:

“We found therapeutic regimen at regulating anti- and protumoral neutrophil phenotypic states by means of four different administration modalities of a TGF- β inhibitor and IFN- β cytokine in lung cancer. The optimal control strategy predicts that, depending on pathological states, therapies may have to be adjusted accordingly to minimize adverse effects of drugs and its administration cost.”

[INTRODUCTION SECTION, page 2, 6th paragraph (whole), in the (marked) revised manuscript]

In the same vein, in order to make Conclusion Section more structured and focused, we added the following paragraph:

“We identified therapeutic regimen for promoting anti-tumoral TANs and suppressing protumoral neutrophil phenotypic states through four different administration of the TGF- β inhibitor and IFN- β in lung cancer. The optimal control scheme predicts that, depending on relative states of N1 and N2 phenotypes, therapy schedules may have to be adjusted properly to minimize adverse effects of these drugs and its administration cost in addition to maximizing anti-tumor efficacy. ”

[page 17, CONCLUSION SECTION, 1st paragraph, lines 10 - 15, in the (marked) revised manuscript]

2. Since TGF- β inhibitor and IFN- β affect a broader range of cell types in the tumor microenvironment other than the TAN, Treg, CD8+, and Th17 cells that considered in the mathematical model, the simulations in this study may be focused on the limited effects of TGF- β inhibitor and IFN- β in the regulation of tumor growth. Of course, it is difficult to take all factors into account in a mathematical model. Still, the limitation should be clarified, for example, that this mathematical model ignores the effects of TGF- β inhibitor and IFN- β on NK cells, tumor-associated fibroblasts, and so on.

(Response) Thank you for careful reading and suggestions. We now organized limitations in three items with more detailed discussions on limitations of our work. In particular, we included detailed discussions on effect of TGF- β inhibitor and IFN- β on NK cells and tumor-associated fibroblasts. So, we replaced

“It has been reported that a more a more realistic clinical situation employs linear cost functions. These type of problems have been used in several works devoted to the administration of single and combination therapies to treat different types of tumors [19, 34, 50, 51, 55]. On the contrary, a study by Glick and Mastroberardino [15] concluded that quadratic control yields continuous, low doses of the therapeutic drug producing a better

outcome in the eradication of the solid tumor. Several researches on optimal control approaches for cancer treatment still favor quadratic controls [41, 52, 58]. Under certain circumstances, both linear and quadratic controls obtain qualitatively similar results [9, 12, 33]. It is true that profiles of treatment strategies depend on the landscape of the cost functionals. These differences show the importance of carefully defining an objective functional that most accurately reflects the toxicities of a particular drug along with the objective of the treatment strategy, for instance, decreasing the tumor mass at the end of the treatment interval, reducing the overall tumor burden over the treatment interval, and some other clinically relevant criteria [14]. It is therefore suggested that further model iterations should include exhaustive investigations on linear controls to have a holistic treatment strategies for cancer treatment. Further analysis of models with other types of objective functionals should also be pursued. Varying results should then be presented to the medical practitioner for them to decide which solutions best fit the biological situation and can be used for practical implementation, if any [33].

The current work does not consider the role of other factors in TME such as intracellular pathways such as STAT1/STAT3 in response to IFN- β , TGF- β , and other stimuli [18], neutrophil elastase (NE) in the presence [20] and absence [43, 16] of LPS, tumor-promoting or -suppressive immune response of NK cells [32, 1], ECM remodeling [11, 35, 27, 29], angiogenesis from blood vessels [49, 48, 61], or growth factors such as EGF [59, 40], and CSF-1 [26, 47]. NK cells were shown to mediate dual roles of neutrophils in metastatic colonization [38, 60]. For example, neutrophils can enhance extravasation of circulating tumor cells by suppressing NK cells [60]. Depletion or adjuvant therapy of NK cells was shown to increase anti-tumor efficacy in a combination therapy (OV-bortezomib-NK) relative to control, showing nonlinear behavior of immune system [32]. These factors may play major roles in regulation of N1 \rightarrow N2 transition, thus cancer progression. For example, NE and NET were shown to promote tumor growth and invasion [37] by turning on the multiple signaling pathways including PI3K in lung cancer cells [24]. In particular, up-regulated NET activities near tumour sections induce the transformation of B cells [54], contributing to cancer progression [65]. However, NET was also suggested to suppress tumor growth in colonic adenocarcinoma [2]. These factors and unforeseen microenvironmental factors may limit bi-lateral switches between N1 and N2 TANs. In our study, optimal control was applied only to IFN- β and TGF- β inhibitors. We plan to investigate the specific role and optimization of these stimuli in TME for better understanding of role of TANs (either promotion [7, 10, 66, 64] or suppression of tumor progression [2]). A new optimal control method has to be developed for a possible triple combination therapy, *i.e.*, TGF- β +TGF- β inhibitor+immune agents [3]. This work, however, provides a general framework of optimal control approach for the fundamental transition from N2 and N1 TANs, thus inhibiting tu-

mor growth, in response to known key players. We plan to develop an optimal control of the TAN's plasticity of the possibly continuous spectrum of the $N1 \rightarrow N2$ switches, which requires better understanding and experiments of the biological system as well as advanced optimal control theory."

with

"Our study has four limitations:

- The current work does not consider the role of other factors in TME such as intracellular pathways, for instance, STAT1/STAT3/JAK [36] in response to IFN- β , TGF- β , and other stimuli [18], neutrophil elastase (NE) in the presence [20] and absence [43, 16] of LPS, tumor-promoting or -suppressive immune response of NK cells [32, 1], tumor-associated fibroblasts (TAFs) [69, 53, 31, 30], ECM remodeling [11, 35, 27, 29], angiogenesis from blood vessels [49, 48, 61], or growth factors such as EGF [59, 40], and CSF-1 [26, 47]. NK cells were shown to mediate dual roles of neutrophils in metastatic colonization [38, 60]. For example, neutrophils can enhance extravasation of circulating tumor cells by suppressing NK cells [60]. Depletion or adjuvant therapy of NK cells was shown to increase anti-tumor efficacy in a combination therapy (OV-bortezomib-NK) relative to control, showing nonlinear behavior of immune system [32]. These factors may play major roles in regulation of $N1 \rightarrow N2$ transition, thus cancer progression. In particular, we focused on the effect of TGF- β inhibitor and IFN- β on TANs in this study but it would be important to see the effect of those two inhibitors on NK cells and TAFs in future work. For example, TGF- β and its inhibitor [69, 53, 17] and IFNs [4, 17, 39, 6] play an important role in regulation of tumor-associated fibroblasts in cancer progression. We plan to include those players in a future optimal-control model.
- NET and NE were shown to promote tumor growth and invasion [37] by turning on the multiple signaling pathways including PI3K in lung cancer cells [24]. For example, up-regulated NET activities near tumour sections induce the transformation of B cells [54], contributing to cancer progression [65]. In particular, mathematical models [37] and experimental data [45] suggest that NET can mediate the critical metastatic process [13, 8] to stabilize the circulating tumor cells in the blood stream and help extravasation of these cancerous cells. However, NET was also suggested to suppress tumor growth in colonic adenocarcinoma [2]. In our study, we did not take into account these critical influence of NETs in a spatial domain. We plan to develop an optimal control approach in a new framework of the partial differential equations of NETs in order to improve the therapeutic, anti-invasion strategies.
- Unforeseen microenvironmental factors may limit bi-lateral switches between $N1$ and $N2$ TANs. In our study, optimal control was applied only to IFN- β and TGF- β inhibitors. We plan to investigate the specific

role and optimization of these stimuli in TME for better understanding of role of TANs (either promotion [7, 10, 66, 64] or suppression of tumor progression [2]). A new optimal control method has to be developed for a possible triple combination therapy, *i.e.*, TGF- β +TGF- β inhibitor+immune agents [3]. We plan to develop an optimal control of the TAN's plasticity of the possibly continuous spectrum of the N1 \rightarrow N2 switches, which requires better understanding and experiments of the biological system as well as advanced optimal control theory.

- It has been reported that a more a more realistic clinical situation employs linear cost functions. These type of problems have been used in several works devoted to the administration of single and combination therapies to treat different types of tumors [19, 34, 50, 51, 55]. On the contrary, a study by Glick and Mastroberardino [15] concluded that quadratic control yields continuous, low doses of the therapeutic drug producing a better outcome in the eradication of the solid tumor. Several researches on optimal control approaches for cancer treatment still favor quadratic controls [41, 52, 58]. Under certain circumstances, both linear and quadratic controls obtain qualitatively similar results [9, 12, 33]. It is true that profiles of treatment strategies depend on the landscape of the cost functionals. These differences show the importance of carefully defining an objective functional that most accurately reflects the toxicities of a particular drug along with the objective of the treatment strategy, for instance, decreasing the tumor mass at the end of the treatment interval, reducing the overall tumor burden over the treatment interval, and some other clinically relevant criteria [14]. It is therefore suggested that further model iterations should include exhaustive investigations on linear controls to have a holistic treatment strategies for cancer treatment. Further analysis of models with other types of objective functionals should also be pursued. Varying results should then be presented to the medical practitioner for them to decide which solutions best fit the biological situation and can be used for practical implementation, if any [33].

This work, however, provides a general framework of optimal control approach for the fundamental transition from N2 and N1 TANs, thus inhibiting tumor growth, in response to known key players. ”

[page 17, last paragraph - page 18, 5th paragraph, in the (marked) revised manuscript]

3. The half-life of circulating neutrophils is reported to be 19 hours (Lahoz-Beneytez et al., Blood, 2016). Therefore, this reviewer speculates that all TANs would be replaced in a few days, although the tumor microenvironment may prolong the life of TANs as the authors described in the Discussion. In Figure 5, for example, the N2/N1 ratio is kept around 1.0 under the conditions that IFN- β remains at zero concentration for several days after degradation. However, this reviewer concerns that the N2/N1 ratio

would return to the initial condition during the IFN- β free period due to the turnover of TANs.

(Response) Thank you for pointing out this and thoughtful questions. Yes, the half-life of TANs is short and TME tends to prolong the half-life. In Fig 5, green bar does not represent IFN- β concentration rather it represents the rate of injection of IFN- β . The IFN- β concentration is marked in green curve in Fig 5A. The whole dynamics essentially leads to relatively similar population of N1 (pink curve in Fig 5A) and N2 (blue curve in Fig 5A) TANs in response to the fluctuating levels of IFN- β (concentration green curve in Fig 5A) following governing equations (2). Please note that concentrations of N2 and N1 TANs are in the similar ranges in Fig 5A. Please note that the bifurcation diagrams in Fig 2(D-F) are steady state values of N2 (variable C) and N1 (variable I) when $\frac{dC}{dt} = \frac{dI}{dt} = 0$, not dynamically changing values. As one can see in Fig 2, the N2 and N1 level is somewhere around 1 in Fig 5. And the concentrations (C, I) are on the way to equilibrium when IFN- β supply rate is low as you mentioned. However, it takes time for the system to adapt the stimulus and converge to an equilibrium point and stimulus such as injections of IFN- β can perturb it. The whole dynamical system determines the slow or fast transition between N1 and N2 TANs. Ideally, the very low values of IFN- β (such as zero values in some intervals) will try to induce N1 TANs if there are no other factors but there are other factors such as TGF- β which transforms N1 TANs to N2 TANs. Therefore, we are not in the N1-dominant system in Fig 5. The N2/N1 ratio essentially is determined by this dynamical system in the deterministic system of ordinary differential equations. The half-lives of N1 and N2 TANs are already taken into account in the mathematical model with an appropriate scaling. With the injection rates that we have in hands in Fig 5, the N2/N1 ratio stays around 1.

4. The differential equations (2) consist of 6 variables (L, G, C, S, I, and T). If this reviewer understands correctly, the time-course of L, G, C, S, and I are shown according to the four optimal control scenarios (Figures 3, 5, 7, and 10), but “T” does not appear in any figures. Therefore, the authors should show the time-course of T, how the tumor size is regulated in response to the treatment of TGF- β inhibitor and/or IFN- β .

(Response) Thank you for careful reading and suggestions. Following the reviewer’s suggestion, we now added time courses of T in Fig 4, Fig 6, Fig 9. Fig 12, so that readers can understand how the tumor size is regulated in response to the treatment of TGF- β inhibitor and/or IFN- β . In addition, we updated the captions for Fig 4, 5, 6, 9, 12 and added statements in the main text as follows:

New caption of Fig 4: “(A) Proportion of the amount of TGF- β inhibitor

used for the entire treatment duration, (B) dynamics of the N1-N2 system, and (C) tumor dynamics under TGF- β inhibitor control starting at different initial conditions Q_1, Q_2, Q_3, Q_4 .”

[page 10, caption of Fig 4, in the (marked) revised manuscript]

Added texts in main text for Fig 4: “Time courses of tumor volume in these four cases illustrate the relatively poor anti-tumor efficacy when the initial condition is in the tumorigenic region Q_4 (Figure 4(C)), despite the highest cost (Figure 4(A)).”

[page 9, 1st paragraph, lines 4-6, in the (marked) revised manuscript]

Figure 4: (A) Proportion of the amount of TGF- β inhibitor used for the entire treatment duration, (B) dynamics of the N1-N2 system, and (C) tumor dynamics under TGF- β inhibitor control starting at different initial conditions Q_1, Q_2, Q_3, Q_4 .

New caption of Fig 6: “(A) Proportion of the amount of IFN- β used for the entire treatment duration, (B) dynamics of the N1-N2 system, and (C) tumor dynamics under IFN- β control starting at different initial conditions Q_1, Q_2, Q_3, Q_4 .”

[page 12, caption of Fig 6, in the (marked) revised manuscript]

Added texts in main text for Fig 6: “Relatively efficient treatment results starting from anti-tumorigenic sector Q_2 and poor outcomes starting from tumorigenic status Q_4 are reflected in time courses of tumor volumes in Figure 6(C). ”

[page 11, 2nd paragraph, lines 4-6, in the (marked) revised manuscript]

Figure 6: (A) Proportion of the amount of IFN- β used for the entire treatment duration, (B) dynamics of the N1-N2 system, and (C) tumor dynamics under IFN- β control starting at different initial conditions Q_1, Q_2, Q_3, Q_4 .

New caption of Fig 9: “Proportion of the amount of TGF- β inhibitor (A) and IFN- β (B) used for the entire treatment duration, (C) dynamics of the N1-N2 system, and (D) tumor volume relative to the high-risk state at the end simulation time under concomitant TGF- β inhibitor and IFN- β controls starting at different initial conditions Q_1, Q_2, Q_3, Q_4 . ”

[page 14, caption of Fig 9, in the (marked) revised manuscript]

Added texts in main text for Fig 9: “Scaled tumor volumes (Q_4 bar in Figure 6(D)) and elongated path from the initial TAN distribution in the tumorigenic zone Q_4 (red dotted curve in Figure 6(C)) also indicate the relatively worst outcome in decreasing the tumor size in the presence of a N2-dominant tumor microenvironment. ”

[page 14, 1st paragraph, lines 2-5, in the (marked) revised manuscript]

Figure 9: Proportion of the amount of TGF- β inhibitor (A) and IFN- β (B) used for the entire treatment duration, (C) dynamics of the N1-N2 system, **and (D) tumor volume relative to the high-risk state at the end simulation time** under concomitant TGF- β inhibitor and IFN- β controls starting at different initial conditions Q_1, Q_2, Q_3, Q_4 .

New caption of Fig 12: (A) Proportion of the amount of TGF- β inhibitor, (B) IFN- β used for the entire treatment duration, (C) dynamics of the N1-N2 system, **and (D) tumor dynamics** under alternating TGF- β inhibitor and IFN- β controls starting at different initial conditions Q_1, Q_2, Q_3, Q_4 . [page 16, caption of Fig 12, in the (marked) revised manuscript]

Added texts in main text for Fig 12: “Time courses of tumor volumes in four cases (Figure 12(D)) show the worst outcome in decreasing the tumor size with the initial TAN distribution in the tumorigenic zone Q_4 , indicating the critical role of the N1/N2 immune conditions. ”

Figure 12: (A) Proportion of the amount of TGF- β inhibitor, (B) IFN- β used for the entire treatment duration, (C) dynamics of the N1-N2 system, and (D) tumor dynamics under alternating TGF- β inhibitor and IFN- β controls starting at different initial conditions Q_1, Q_2, Q_3, Q_4 .

I have a few additional minor comments, explained below.

- In the third paragraph of the Introduction, the authors say, TGF- β inhibitor such as gefitinib. However, gefitinib is an EGFR inhibitor, not a TGF- β inhibitor.

(Response) Thank you for careful reading. Indeed, it is an EGF inhibitor. We now replaced “gefitinib [56]” with “galunisertib [22, 23, 68, 57, 67] and LY2109761 [21, 42]” [page 2, 3rd paragraph, line 6, in the (marked) revised manuscript]

- The authors assumed that TGF- β inhibitor and IFN- β can be administrated every other day in the simulations, but what is the evidence?

(Response) Thank you for careful reading. We added the evidences for IFN- β (as well as other references for toxicity) by replacing

“Taking into account that IFN- β is a noxious drug, it can be administered only every other day in increasing amount.”

with

“Taking into account that IFN- β is a noxious drug [62, 25, 63], it can be administered only every other day in increasing amount as illustrated in experimental system [46, 5, 62].”

[page 10, 1st paragraph, lines 2-4, in the (marked) revised manuscript]

In addition to these changes, we made the following minor changes in the (marked) revised manuscript in order to meet the journal requirement (from editorial staffs)

(i) “Ohio, ” \rightarrow “OH 43210, ” [page 1, author affiliation]

REFeree 3:

In the paper the authors consider a mathematical model of tumor-associated neutrophils in lung cancer progression. This model uses a mathematical model to explain the optimal control problems of TGF- β and IFN- β . The part of the mathematical model has been revised in detail in a previous review of another Journal. However, there are some questions about the contents of the paper. Below are these comments:

1. The authors define Q1 to Q4 in the Results and Discussion section, but how do you apply them to individual cases in the actual clinical settings? How do you measure the level of N1 and N2 component? Is it by a blood test or histopathological examination such as surgical specimens or biopsy specimens?

(Response) As we can see from the analysis of the mathematical model, there are relatively high and low probabilities of moving toward the N2-dominant status (high N2 and low N1), i.e, tumorigenic status, depending on the relative proportion of N1 and N2 populations in the given patient as in Fig 4. We acknowledge that it is very challenging to measure the level of N1 and N2 components. The researchers in the field is moving fast for further experiments and etc. For instance, Ohms et al (2020) [44] could measure the relative proportion of N1 and N2 populations. However, it is still challenging to measure and use it in a clinical setting yet. If one can measure individual levels of N2 and N1 in the blood and TME in each patient, not N2/N1 ratio, whether it is by a blood test or histopathological examination such as surgical specimens or biopsy specimens, one can determine their locations (Q1, Q2, Q3, Q4) in the N2-N1 phase plane and apply our strategies for optimal results. As we mentioned in the response to item 2 below, we are vigorously working with experimentalists (professor Jinsu Kim for instance) and clinicians at KIRAMs and Korea Institute of radiological and medical sciences in Seoul, South Korea, in order to develop this particular method to overcome this technical difficulties.

2. Have you verified the results of this mathematical model in actual clinical cases or in vitro? If it can be shown that the results of actual biological research and mathematical models match, I think it will be a more meaningful paper in the clinical settings. However, it does not force the authors to do biological research when it is difficult to research.

(Response) Thank you very much for nice suggestions. It would be great if we can show that the simulated results from the mathematical model can be integrated or verified in a series of experiments or in a clinical setting. Our colleagues at KIRAMs and Korea Institute of radiological and medical sciences (hospital especially for cancer patients at last stage) are actually trying very hard to do experiments to verify and optimize the outcomes of the treatment. However, practically speaking, this type of experiments requires lots of approval and internal processes and it usually takes much time to get patients and setup the program in the clinical setting. We also tried to integrate the simulating tools to the system at the hospital but we have to overcome a large number of hurdles due to government regulations and ethical issues etc. We hope to do it in near future. However, our paper could contribute to better research environment where experimentalists can actively participate for joint projects so that the system with mathematical models can function altogether.

In addition to these changes, we made the following minor changes in the (marked) revised manuscript in order to meet the journal requirement (from editorial staffs)

- (i) **“Ohio, ” → “OH 43210, ” [page 1, author affiliation]**

References

- [1] Aktas, O., Ozturk, A., Erman, B., Erus, S., Tanju, S., and Dilege, S. (2018). Role of natural killer cells in lung cancer. *J Cancer Res Clin Oncol*, 144(6):997–1003.
- [2] Arelaki, S., Arampatzioglou, A., Kambas, K., Papagoras, C., Miltiades, P., Angelidou, I., Mitsios, A., Kotsianidis, I., Skendros, P., Sivridis, E., Maroulakou, I., Giatromanolaki, A., and Ritis, K. (2016). Gradient infiltration of neutrophil extracellular traps in colon cancer and evidence for their involvement in tumour growth. *PLoS One*, 11(5):e0154484.
- [3] Aspirin, A. P., de los Reyes V, A. A., and Kim, Y. (2020). Polytherapeutic strategies with oncolytic virus-bortezomib and adjuvant NK cells in cancer treatment. *Journal of the Royal Society Interface*, in press.
- [4] Broad, R., Jones, S., Teske, M., Wastall, L., Hanby, A., Thorne, J., and Hughes, T. (2021). Inhibition of interferon-signalling halts cancer-associated fibroblast-dependent protection of breast cancer cells from chemotherapy. *Br J Cancer*, 124(6):1110–1120.

- [5] Catani, J. P. P., Medrano, R. F. V., Hunger, A., Del Valle, P., Adjemian, S., Zanatta, D. B., Kroemer, G., Costanzi-Strauss, E., and Strauss, B. E. (2016). Intratumoral immunization by p19arf and interferon- β gene transfer in a heterotopic mouse model of lung carcinoma. *Translational oncology*, 9(6):565–574.
- [6] Cho, C., Mukherjee, R., Peck, A., Sun, Y., McBrearty, N., Katlinski, K., Gui, J., Govindaraju, P., Pure, E., Rui, H., and Fuchs, S. (2020). Cancer-associated fibroblasts down-regulate type i interferon receptor to stimulate intratumoral stromagenesis. *Oncogene*, 39(38):6129–6137.
- [7] Cools-Lartigue, J., Spicer, J., Najmeh, S., and Ferri, L. (2014). Neutrophil extracellular traps in cancer progression. *Cell Mol Life Sci*, 71(21):4179–94.
- [8] Das, A., Monteiro, M., Barai, A., Kumar, S., and Sen, S. (2017). Mmp proteolytic activity regulates cancer invasiveness by modulating integrins. *Sci Rep*, 7(1):14219.
- [9] de Pillis, L., Gu, W., Fister, K., Head, T., Maples, K., Murugan, A., Neal, T., and Yoshida, K. (2007). Chemotherapy for tumors: An analysis of the dynamics and a study of quadratic and linear optimal controls. *Mathematical Biosciences*, 209(1):292–315.
- [10] Demers, M. and Wagner, D. (2013). Neutrophil extracellular traps: A new link to cancer-associated thrombosis and potential implications for tumor progression. *Oncoimmunology*, 2(2):e22946.
- [11] Eisenberg, M., Kim, Y., Li, R., Ackerman, W., Kniss, D., and Friedman, A. (2011). Modeling the effects of myoferlin on tumor cell invasion. *Proc Natl Acad Sci USA*, 108(50):20078–83.
- [12] Elmouki, I. and Saadi, S. (2016). Quadratic and linear controls developing an optimal treatment for the use of bcg immunotherapy in superficial bladder cancer. *Optimal Control Applications and Methods*, 37(1):176–189.
- [13] Fares, J., Fares, M., Khachfe, H., Salhab, H., and Fares, Y. (2020). Molecular principles of metastasis: A hallmark of cancer revisited. *Signal Transduct Target Ther*, 5(1):28.
- [14] Fister, K. R. and Panetta, J. C. (2003). Optimal control applied to competing chemotherapeutic cell-kill strategies. *SIAM Journal on Applied Mathematics*, 63(6):1954–1971.
- [15] Glick, A. E. and Mastroberardino, A. (2017). An optimal control approach for the treatment of solid tumors with angiogenesis inhibitors. *Mathematics*, 5(4):49.
- [16] Gong, L., Cumpian, A., Caetano, M., Ochoa, C., la Garza, M. D., Lapid, D., Mirabol-fathinejad, S., Dickey, B., Zhou, Q., and Moghaddam, S. (2013). Promoting effect of neutrophils on lung tumorigenesis is mediated by cxcr2 and neutrophil elastase. *Mol Cancer*, 12(1):154.

- [17] Grauel, A., Nguyen, B., Ruddy, D., Laszewski, T., Schwartz, S., Chang, J., Chen, J., Piquet, M., Pelletier, M., Yan, Z., Kirkpatrick, N., Wu, J., deWeck, A., Riestler, M., Hims, M., Geyer, F., Wagner, J., MacIsaac, K., Deeds, J., Diwanji, R., Jayaraman, P., Yu, Y., Simmons, Q., Weng, S., Raza, A., Minie, B., Dostalek, M., Chikkegowda, P., Ruda, V., Iartchouk, O., Chen, N., Thierry, R., Zhou, J., Pruteanu-Malinici, I., Fabre, C., Engelman, J., Dranoff, G., and Cremasco, V. (2020). TGF β -blockade uncovers stromal plasticity in tumors by revealing the existence of a subset of interferon-licensed fibroblasts. *Nat Commun*, 11(1):6315.
- [18] Grimes, M., Hall, B., Foltz, L., Levy, T., Rikova, K., Gaiser, J., Cook, W., Smirnova, E., Wheeler, T., Clark, N., Lachmann, A., Zhang, B., Hornbeck, P., Maayan, A., and Comb, M. (2018). Integration of protein phosphorylation, acetylation, and methylation data sets to outline lung cancer signaling networks. *Sci Signal.*, 11(531):pii: eaaq1087.
- [19] Gllmann, L. and Maurer, H. (2018). Optimal control problems with time delays: Two case studies in biomedicine. *Mathematical Biosciences and Engineering*, 15(5):1137–1154.
- [20] Hattar, K., Franz, K., Ludwig, M., Sibelius, U., Wilhelm, J., Lohmeyer, J., Savai, R., Subtil, F., Dahlem, G., Eul, B., Seeger, W., Grimminger, F., and Grandel, U. (2014). Interactions between neutrophils and non-small cell lung cancer cells: enhancement of tumor proliferation and inflammatory mediator synthesis. *Cancer Immunol Immunother*, 63(12):1297–306.
- [21] He, X., Guo, X., Zhang, H., Kong, X., Yang, F., and Zheng, C. (2017). Mechanism of action and efficacy of LY2109761, a TGF- β receptor inhibitor, targeting tumor microenvironment in liver cancer after TACE. *Oncotarget.*, 9(1):1130–1142.
- [22] Herbertz, S., Sawyer, J. S., Stauber, A. J., Gueorguieva, I., Driscoll, K. E., Estrem, S. T., Cleverly, A. L., Desaiyah, D., Guba, S. C., Benhadji, K. A., Slapak, C. A., and Lahn, M. M. (2015). Clinical development of galunisertib (LY2157299 monohydrate), a small molecule inhibitor of transforming growth factor-beta signaling pathway. *Drug Design, Development and Therapy*, 9:4479–4499.
- [23] Holmgaard, R., Schaer, D., and S.P. Castaneda, Y. L., Murphy, M., Xu, X., Inigo, I., Dobkin, J., Manro, J., Iversen, P., Surguladze, D., Hall, G., Novosiadly, R., Benhadji, K., Plowman, G., Kalos, M., and Driscoll, K. (2018). Targeting the TGF β pathway with galunisertib, a TGF β RI small molecule inhibitor, promotes anti-tumor immunity leading to durable, complete responses, as monotherapy and in combination with checkpoint blockade. *J Immunother Cancer.*, 6(1):47.
- [24] Houghton, A., Rzymkiewicz, D., Ji, H., Gregory, A., Egea, E., Metz, H., Stolz, D., Land, S., Marconcini, L., Kliment, C., Jenkins, K., Beaulieu, K., Mouded, M., Frank, S., Wong, K., and Shapiro, S. (2010). Neutrophil elastase-mediated degradation of irs-1 accelerates lung tumor growth. *Nat Med*, 16(2):219–223.
- [25] Jonasch, E. and Haluska, F. (2001). Interferon in oncological practice: review of interferon biology, clinical applications, and toxicities. *Oncologist*, 6(1):34–55.

- [26] Kim, Y., Jeon, H., and Othmer, H. (2017). The role of the tumor microenvironment in glioblastoma: A mathematical model. *IEEE Transactions on Biomedical Engineering*, 64(3):519–527.
- [27] Kim, Y., Lawler, S., Nowicki, M., Chiocca, E., and Friedman, A. (2009). A mathematical model of brain tumor : pattern formation of glioma cells outside the tumor spheroid core. *J. Theo. Biol.*, 260:359–371.
- [28] Kim, Y., Lee, D., Lee, J., Lee, S., and Lawler, S. (2019). Role of tumor-associated neutrophils in regulation of tumor growth in lung cancer development: A mathematical model. *PLoS ONE*, 14(1):1–40.
- [29] Kim, Y. and Othmer, H. (2013). A hybrid model of tumor-stromal interactions in breast cancer. *Bull Math Biol*, 75:1304–1350.
- [30] Kim, Y., Stolarska, M., and Othmer, H. (2011). The role of the microenvironment in tumor growth and invasion. *Prog Biophys Mol Biol*, 106:353–379.
- [31] Kim, Y., Wallace, J., Li, F., Ostrowski, M., and Friedman, A. (2010). Transformed epithelial cells and fibroblasts/myofibroblasts interaction in breast tumor: a mathematical model and experiments. *J. Math. Biol.*, 61(3):401–421.
- [32] Kim, Y., Yoo, J. Y., Lee, T. J., Liu, J., Yu, J., Caligiuri, M. A., Kaur, B., and Friedman, A. (2018). Complex role of NK cells in regulation of oncolytic virus-bortezomib therapy. *Proceedings of the National Academy of Sciences*, 115(19):4927–4932.
- [33] Ledzewicz, U., Brown, T., and Schttler, H. (2004). Comparison of optimal controls for a model in cancer chemotherapy with L1- and L2-type objectives. *Optimization Methods and Software*, 19(3-4):339–350.
- [34] Ledzewicz, U., Wang, S., Schttler, H., Andr, N., Heng, M. A., and Pasquier, E. (2017). On drug resistance and metronomic chemotherapy: A mathematical modeling and optimal control approach. *Mathematical Biosciences and Engineering*, 4(1):217–235.
- [35] Lee, B., Konen, J., Wilkinson, S., Marcus, A., and Jiang, Y. (2017). Local alignment vectors reveal cancer cell-induced ecm fiber remodeling dynamics. *Sci Rep*, 7:39498.
- [36] Lee, J., Lee, D., and Kim, Y. (2021a). Mathematical model of stat signalling pathways in cancer development and optimal control approaches. *Royal Soc. Open Sci.*, 8:210594.
- [37] Lee, J., Lee, D., Lawler, S., and Kim, Y. (2021b). Role of neutrophil extracellular traps in regulation of lung cancer invasion and metastasis: Structural insights from a computational model. *PLOS Computational Biology*, 17(2):e1008257.
- [38] Li, P., Lu, M., Shi, J., Hua, L., Gong, Z., Li, Q., Shultz, L., and Ren, G. (2020). Dual roles of neutrophils in metastatic colonization are governed by the host nk cell status. *Nat Commun*, 11(1):4387.

- [39] Liu, T., Han, C., Wang, S., Fang, P., Ma, Z., Xu, L., and Yin, R. (2019). Cancer-associated fibroblasts: an emerging target of anti-cancer immunotherapy. *J Hematol Oncol*, 12(1):86.
- [40] Liu, X., Wang, W., Zhang, C., and Ma, Z. (2017). Epidermal growth factor receptor (EGFR): A rising star in the era of precision medicine of lung cancer. *Oncotarget*, 8(30):50209–50220.
- [41] Malinzi, J., Ouifki, R., Eladdadi, A., Torres, D. F. M., and White, K. A. J. (2018). Enhancement of chemotherapy using oncolytic virotherapy: Mathematical and optimal control analysis. *Mathematical Biosciences & Engineering*, 15(6):1435–1463.
- [42] Melisi, D., Ishiyama, S., Scwabas, G. M., Fleming, J. B., Xia, Q., Tortora, G., Abbruzzese, J. L., and Chiao, P. J. (2008). Ly2109761, a novel transforming growth factor β receptor type i and type ii dual inhibitor, as a therapeutic approach to suppressing pancreatic cancer metastasis. *Molecular cancer therapeutics*, 7(4):829–840.
- [43] Moroy, G., Alix, A., Sapi, J., Hornebeck, W., and Bourguet, E. (2012). Neutrophil elastase as a target in lung cancer. *Anticancer Agents Med Chem*, 12(6):565–79.
- [44] Ohms, M., Moller, S., and Laskay, T. (2020). An attempt to polarize human neutrophils toward n1 and n2 phenotypes in vitro. *Front Immunol*, 11:532.
- [45] Park, J., Wysocki, R., Amoozgar, Z., Maiorino, L., Fein, M., Jorns, J., Schott, A., Kinugasa-Katayama, Y., Lee, Y., Won, N., Nakasone, E., Hearn, S., Kuttner, V., Qiu, J., Almeida, A., Perurena, N., Kessenbrock, K., Goldberg, M., and Egeblad, M. (2016). Cancer cells induce metastasis-supporting neutrophil extracellular dna traps. *Sci Transl Med*, 8(361):361ra138.
- [46] Patel, M., Jacobson, B., Ji, Y., Drees, J., Tang, S., Xiong, K., Wang, H., Prigge, J., Dash, A., Kratzke, A., Mesev, E., Etchison, R., Federspiel, M., Russell, S., and Kratzke, R. (2015). Vesicular stomatitis virus expressing interferon- β is oncolytic and promotes antitumor immune responses in a syngeneic murine model of non-small cell lung cancer. *Oncotarget*, 6(32):33165–77.
- [47] Pyonteck, S., Akkari, L., Schuhmacher, A., Bowman, R., Sevenich, L., Quail, D., Olson, O., Quick, M., Huse, J., Teijeiro, V., Setty, M., Leslie, C., Oei, Y., Pedraza, A., Zhang, J., Brennan, C., Sutton, J., Holland, E., Daniel, D., and Joyce, J. (2013). CSF-1R inhibition alters macrophage polarization and blocks glioma progression. *Nat Med*, 19(10):1264–72.
- [48] Qu, J., Zhang, Y., Chen, X., Yang, H., Zhou, C., and Yang, N. (2017). Newly developed anti-angiogenic therapy in non-small cell lung cancer. *Oncotarget*, 9(11):10147–10163.
- [49] Revels, S. and Lee, J. (2018). Anti-angiogenic therapy in nonsquamous non-small cell lung cancer (NSCLC) with tyrosine kinase inhibition (TKI) that targets the VEGF receptor (VEGFR): perspective on phase III clinical trials. *J Thorac Dis*, 10(2):617–620.

- [50] Rihan, F., Lakshmanan, S., and Maurer, H. (2019). Optimal control of tumour-immune model with time-delay and immuno-chemotherapy. *Applied Mathematics and Computation*, 353:147–165.
- [51] Rodriguez, C. R., Fernandez Calvo, G., Ramis-Conde, I., and Belmonte-Beitia, J. (2017). Stochastic modelling of slow-progressing tumors: Analysis and applications to the cell interplay and control of low grade gliomas. *Communications in Nonlinear Science and Numerical Simulation*, 49:63–80.
- [52] Sabir, S., Raissi, N., and Serhani, M. (2020). Chemotherapy and immunotherapy for tumors: A study of quadratic optimal control. *International Journal of Applied and Computational Mathematics*, 6(3):81.
- [53] Sahai, E., Astsaturov, I., Cukierman, E., DeNardo, D., Egeblad, M., Evans, R., Fearon, D., Greten, F., Hingorani, S., Hunter, T., Hynes, R., Jain, R., Janowitz, T., Jorgensen, C., Kimmelman, A., Kolonin, M., Maki, R., Powers, R., Pure, E., Ramirez, D., Scherz-Shouval, R., Sherman, M., Stewart, S., Tlsty, T., Tuveson, D., Watt, F., Weaver, V., Weeraratna, A., and Werb, Z. (2020). A framework for advancing our understanding of cancer-associated fibroblasts. *Nat Rev Cancer*, 20(3):174–186.
- [54] Sangaletti, S., Tripodo, C., Vitali, C., Portararo, P., Guarnotta, C., Casalini, P., Cappetti, B., Miotti, S., Pinciroli, P., Fuligni, F., Fais, F., Piccaluga, P., and Colombo, M. (2014). Defective stromal remodeling and neutrophil extracellular traps in lymphoid tissues favor the transition from autoimmunity to lymphoma. *Cancer Discov.*, 4(1):110–129.
- [55] Schättler, H. and Ledzewicz, U. (2015). Optimal control for mathematical models of cancer therapies. In *An Application of Geometric Methods*. Springer, New York, NY.
- [56] Serizawa, M., Takahashi, T., Yamamoto, N., and Koh, Y. (2013). Combined treatment with erlotinib and a transforming growth factor- β type i receptor inhibitor effectively suppresses the enhanced motility of erlotinib-resistant nonsmall-cell lung cancer cells. *Journal of Thoracic Oncology*, 8(3):259–269.
- [57] Serova, M., Tijeras-Raballand, A., Santos, C., Albuquerque, M., Paradis, V., Neuzillet, C., Benhadji, K., Raymond, E., Faivre, S., and de Gramont, A. (2015). Effects of TGF-beta signalling inhibition with galunisertib (ly2157299) in hepatocellular carcinoma models and in ex vivo whole tumor tissue samples from patients. *Oncotarget.*, 6(25):21614–21627.
- [58] Sharma, S. and Samanta, G. P. (2016). Analysis of the dynamics of a tumor-immune system with chemotherapy and immunotherapy and quadratic optimal control. *Differential Equations and Dynamical Systems*, 24(2):149–171.
- [59] Siegelin, M. and Borczuk, A. (2014). Epidermal growth factor receptor mutations in lung adenocarcinoma. *Lab Invest*, 94(2):129–137.
- [60] Spiegel, A., Brooks, M., Houshyar, S., Reinhardt, F., Ardolino, M., Fessler, E., Chen, M., Krall, J., DeCock, J., Zervantonakis, I., Iannello, A., Iwamoto, Y., Cortez-Retamozo, V., Kamm, R., Pittet, M., Raulet, D., and Weinberg, R. (2016). Neutrophils suppress

- intraluminal nk cell-mediated tumor cell clearance and enhance extravasation of disseminated carcinoma cells. *Cancer Discov*, 6(6):630–49.
- [61] Stratigos, M., Matikas, A., Voutsina, A., Mavroudis, D., and Georgoulas, V. (2016). Targeting angiogenesis in small cell lung cancer. *Transl Lung Cancer Res*, 5(4):389–400.
- [62] Studeny, M., Marini, F. C., Dembinski, J. L., Zompetta, C., Cabreira-Hansen, M., Bekele, B. N., Champlin, R. E., and Andreeff, M. (2004). Mesenchymal Stem Cells: Potential Precursors for Tumor Stroma and Targeted-Delivery Vehicles for Anticancer Agents. *JNCI: Journal of the National Cancer Institute*, 96(21):1593–1603.
- [63] Tavakoli, M., Manshadi, S. P., Naderi, N., Dehghan, A., and Azizi, S. (2012). Unusual side effects of interferon beta-1a in patient with multiple sclerosis. *Mater Sociomed*, 24(3):203–5.
- [64] Tohme, S., Yazdani, H., Al-Khafaji, A., Chidi, A., Loughran, P., Mowen, K., Wang, Y., Simmons, R., Huang, H., and Tsung, A. (2016). Neutrophil extracellular traps promote the development and progression of liver metastases after surgical stress. *Cancer Res*, 76(6):1367–1380.
- [65] Uribe-Querol, E. and Rosales, C. (2015). Neutrophils in cancer: Two sides of the same coin. *J Immunol Res*, 2015::983698.
- [66] Wculek, S. and Malanchi, I. (2015). Neutrophils support lung colonization of metastasis-initiating breast cancer cells. *Nature*, 528(7582):413–417.
- [67] Wick, A., Desjardins, A., Suarez, C., Forsyth, P., Gueorguieva, I., Burkholder, T., Cleverly, A., Estrem, S., Wang, S., Lahn, M., Guba, S., Capper, D., and Rodon, J. (2020). Phase 1b/2a study of galunisertib, a small molecule inhibitor of transforming growth factor-beta receptor i, in combination with standard temozolomide-based radiochemotherapy in patients with newly diagnosed malignant glioma. *Invest New Drugs.*, 38(5):1570–1579.
- [68] Yingling, J., McMillen, W., Yan, L., Huang, H., Sawyer, J., Graff, J., Clawson, D., Britt, K., Anderson, B., Beight, D., Desai, D., Lahn, M., Benhadji, K., Lallena, M., Holmgaard, R., Xu, X., Zhang, F., Manro, J., Iversen, P., Iyer, C., Brekken, R., Kalos, M., and Driscoll, K. (2017). Preclinical assessment of galunisertib (LY2157299 monohydrate), a first-in-class transforming growth factor- β receptor type i inhibitor. *Oncotarget.*, 9(6):6659–6677.
- [69] Yoon, H., Tang, C., Banerjee, S., Delgado, A., Yebra, M., Davis, J., and Sicklick, J. (2021). TGF-beta-mediated transition of resident fibroblasts to cancer-associated fibroblasts promotes cancer metastasis in gastrointestinal stromal tumor. *Oncogenesis*, 10(2):13.